

# Validation of Aeolus L2B products over the tropical Atlantic using radiosondes

Maurus Borne [1], Peter Knippertz [1], Martin Weissmann [2], Benjamin Witschas [3], Cyrille Flamant [4], Rosimar Rios-Berrios [5], and Peter Veals [6]

[1]Institute of Meteorology and Climate Research, Karlsruhe Institute of Technology (KIT), 76131 Karlsruhe, Germany
[2]Universität Wien, Institut für Meteorologie und Geophysik, Wien, Austria
[3]Deutsches Zentrum für Luft- und Raumfahrt e.V. (DLR), Institut für Physik der Atmosphäre, 82234 Oberpfaffenhofen, Germany
[4]Laboratoire Atmosphères, Milieux, Observations Spatiales (LATMOS), UMR 8190, CNRS, Sorbonne Université and Université Paris Saclay, Paris, France
[5]National Center for Atmospheric Research, Boulder, CO, USA
[6]Department of Atmospheric Sciences, University of Utah, Salt Lake City, Utah, USA

**Correspondence:** Maurus Borne (maurus.borne@kit.edu)

**Abstract.** Since its launch by the European Space Agency in 2018, the Aeolus satellite has been using the first Doppler wind lidar in space to acquire three-dimensional atmospheric wind profiles around the globe. Especially in the tropics, these measurements compensate for the currently limited number of other wind observations, making an assessment of the quality of Aeolus wind products in this region crucial for numerical weather prediction. To evaluate the quality of the Aeolus L2B wind products across the tropical Atlantic Ocean, 20 radiosondes corresponding to Aeolus overpasses were launched from the islands of Sal, Saint Croix and Puerto Rico during August-September 2021 as part of the Joint Aeolus Tropical Atlantic Campaign. During this period, Aeolus sampled winds within a complex environment with a variety of cloud types in the vicinity of the Inter-tropical Convergence Zone and aerosol particles from Saharan dust outbreaks. On average, the validation for Aeolus Raleigh-clear revealed a random error of 3.8 – 4.3 ms$^{-1}$ between 2–16 km and 4.3 – 4.8 ms$^{-1}$ between 16–20 km, with a systematic error of -0.5±0.2 ms$^{-1}$. For Mie-cloudy, the random error between 2–16 km is 1.1 – 2.3 ms$^{-1}$ and the systematic error is -0.9 ±0.3 ms$^{-1}$. Below clouds or within dust layers, the quality of Rayleigh-clear measurements can be degraded when the useful signal is reduced. In these conditions, we also noticed an underestimation of the L2B estimated error. Gross outliers which we define with large deviations from the radiosonde but low error estimates account for less than 5% of the data. These outliers appear at all altitudes and under all environmental conditions; however, their root-cause remains unknown. Finally, we confirm the presence of an orbital-dependent bias of up to 2.5 ms$^{-1}$ observed with both radiosondes and European Centre for Medium-Range Weather Forecasts model equivalents. The results of this study contribute to a better characterization of the Aeolus wind product in different atmospheric conditions and provide valuable information for further improvement of the wind retrieval algorithm.





# 1   Introduction

According to the World Meteorological Organisation (WMO), wind is the most critical atmospheric variable lacking in the current Global Observing System (GOS) (Baker et al., 2014). Especially in the Southern Hemisphere (SH), over the oceans and near equatorial regions, numerical weather prediction (NWP) models require additional wind observations with sufficient coverage in time and space to identify key atmospheric dynamics (Stoffelen et al., 2005; Straume et al., 2020). Before the launch of Aeolus in 2018, satellite wind observations in these regions were only available for a limited number of tropospheric layers and were mainly provided by atmospheric motion vectors (AMVs) estimated from tracking cloud and water vapour features (Bormann et al., 2003; Folger and Weissmann, 2014), or by scatterometer measurements of surface winds (Naderi et al., 1991; Portabella and Stoffelen, 2009). In situ measurements derived from aircraft reports, ground stations or radiosondes are not globally distributed and lead to a lack of observations in the aforementioned regions.

To address these deficiencies, the European Space Agency (ESA) deployed the Atmospheric Dynamics Mission Aeolus in 2018, the first satellite capable of measuring atmospheric winds around the globe from space with a homogeneous space-time wind coverage and altitude-resolved profiles up to 30 km height (Reitebuch, 2012). The instrument carries a direct detection Doppler wind lidar called ALADIN (Atmospheric LAser Doppler INstrument) that emits short ultraviolet (UV) pulses at 355 nm along the Line Of Sight (LOS) of the instrument. The Doppler shift of the backscatter signal is detected by a dual-channel receiver consisting of the following elements: a Fizeau interferometer analysing the Doppler shift of the narrowband particle backscatter signal (cloud droplets and aerosols or ice crystals) using the fringe imaging technique (McKay, 2002), referred to as the Mie channel, and a dual Fabry-Pérot interferometer detecting the Doppler-shifted frequency of the Rayleigh-Brillouin backscatter spectrum (air molecules) using the double-edge technique (Flesia and Korb, 1999), called the Rayleigh channel. The processing algorithm also distinguishes between retrievals originating from "cloudy" or "clear" atmospheric conditions, resulting in Rayleigh-clear and Mie-cloudy observation types. The two channels complement each other, as Mie-cloudy winds can compensate for gaps in Rayleigh-clear measurements, especially in cloudy and aerosol-loaded regions. Various NWP centres have demonstrated the added value of assimilating Aeolus winds through significant improvements in model fields and model background information, especially in tropical regions, the upper tropical troposphere and the lower stratosphere (Rennie et al., 2021; Martin et al., 2022a, b; Garrett et al., 2022).

For an optimal use of the Aeolus wind observations in NWP models, an assessment of the data quality is essential. To achieve this, several scientific and technical studies are carried out in the framework of Calibration/Validation (Cal/Val) activities organised by ESA. For wind validation, several reference products have been used such as ground-based remote sensing observations (Belova et al., 2021; Guo et al., 2021; Iwai et al., 2021; Abril-Gago et al., 2022), in situ measurements (Baars et al., 2020; Chen et al., 2021; Ratynski et al., 2022), airborne measurements (Lux et al., 2020; Witschas et al., 2020; Bedka et al., 2021; Witschas et al., 2022) or NWP model equivalents (Martin et al., 2021; Zuo et al., 2022).

Several anomalies in the Aeolus data have already been detected and improvements in the processing chain and the instrument have been made accordingly. These include the implementation of a bias correction in both channels related to the orbital-dependent temperature variations of ALADIN's M1 mirror (Weiler et al., 2021b) and the correction of "pixel anoma-



lies" (Weiler et al., 2021a), which reduced the systematic and random errors in the Rayleigh channel. One phenomenon that still needs to be explored is the sensitivity of Aeolus wind quality to the presence of aerosols and clouds, potentially affecting

key parameters such as the signal levels or scattering ratio (SR) used to calculate the Horizontal Line of Sight (HLOS) winds and the associated error estimate (EE). The tropical Atlantic during the boreal summer, spanning from West Africa to the Caribbean, is the ideal place to explore these dependencies, with a wide range of atmospheric aerosols (Saharan dust aerosols, sea salt aerosols, biomass combustion aerosols) and convective cloud types associated with the West African Monsoon (WAM) circulation and the Inter Tropical Convergence Zone (ITCZ).

For this purpose, ESA organized the Joint Aeolus Tropical Atlantic Campaign (JATAC) in the period July to September 2021, which deployed sophisticated airborne lidar instruments over Cabo Verde (German Aerospace Center (DLR), Laboratoire ATmosphères, Milieux, Observations Spatiales (LATMOS)) and the Virgin Islands (National Aeronautics and Space Administration (NASA)) but also ground-based instruments such as radiosondes (Karlsruhe Institute of Technology (KIT), University of Oklahoma, University of Utah) and Doppler lidar systems (Leibniz Institute for Tropospheric Research (TROPOS), National Observatory of Athens (NOA)). In this study, we validate Aeolus wind products using radiosondes launched

from western Puerto Rico, northern St. Croix and Sal airport on Cabo Verde. The semi-arid island of Sal is located over the tropical East Atlantic off the West African coast, near the northern boundary of the WAM. Rain events are relatively sporadic there, as most synoptic and mesoscale precipitation systems propagate south of the island. The region is exposed to mineral dust plumes emanating from Saharan dust outbreaks. In contrast, the islands of St. Croix and Puerto Rico are located within the warm and moist Caribbean, where heavy rainfall events and tropical cyclones frequently affect the area. The contribution

of the radiosondes in JATAC is complementary to other instruments as they provide accurate wind measurements throughout the troposphere up to the lower stratosphere, which is not probed by many other instruments and provides an almost unique data set for validating the Aeolus winds at this altitude.

This article is structured as follows: Section 2 describes the instruments and data while section 3 details the quality control and co-location criteria used for the validation study. Section 4 deals with the quantification of errors, their dependency

on temporal and spatial distance between the compared observations as well as on the presence of clouds and dust. For this purpose, we use the Satellite Application Facility for supporting NoWCasting and very short range forecasting (SAFNWC, Alonso Lasheras et al. (2005)) satellite-based meteorological Cloud Type (CT) product and the Copernicus Atmosphere Monitoring Service (CAMS) dust mixing ratio reanalysis. Furthermore, the section includes a case study illustrating the different

behaviour of Rayleigh-clear and Mie-cloudy winds under different environmental conditions. Finally, section 5 summarises the main results and provides recommendations for improving the Aeolus wind retrieval algorithm.

## 2 Instruments and Data

### 2.1 ALADIN and Aeolus wind products

Aeolus is the second Earth Explorer Core mission and measures global atmospheric wind profiles from a 320 km high sun-

synchronous dusk-dawn orbit. It carries the ALADIN instrument (Schillinger et al. (2003)), which is a direct-detection high-



spectral-resolution wind lidar with a Nd:YAG laser transmitter that operates at an ultraviolet wavelength of 354.8 nm. It points at 35 ° with and angle of ∼10° from the zonal direction.

ALADIN consists of a two-channel receiver that allows the instrument to measure wind speed from molecular backscatter (Rayleigh channel) and particle backscatter (Mie channel). The Rayleigh channel relies on the double-edge technique (Flesia and Korb, 1999) using a sequential Fabry-Perot interferometer, where the Doppler shift of the backscattered molecular spectrum is retrieved from the signal intensities that are transmitted through two band-pass filters A and B. The final Rayleigh response is computed from a contrast-function between both filter signals. For Mie winds, the computation is based on a fringe-imaging technique (McKay, 2002), in which the Fizeau interferometer forms a linear interference fringe on the detector from the narrowband particle backscatter signal. The lateral displacement of the interference fringe is then used to calculate the Doppler shift.

To ensure a sufficient Signal-to-Noise Ratio (SNR), the wind measurements are averaged vertically and horizontally into single observations. Vertical sampling is performed within 24 vertical elevation bins with a resolution that can vary from 0.25 km at lower elevations to 2 km at higher elevations. They are defined by the Range Bin Settings (RBS) and can vary geographically and between the respective detection channel (Rayleigh and Mie). Horizontally, the measurements are averaged over 87 km and 10 km integration lengths for Rayleigh and Mie channels, respectively, owing to the lower signal levels of the Rayleigh measurements.

The data products are processed through a multi-stage processing chain, with each level containing different information (Reitebuch et al., 2018; Tan et al., 2008). In this study, the Level1B (L1B) and Level2B (L2B) products are of particular interest. The L1B product comprises the geolocated and observation data as well as optical information (SNR, useful signal, scattering ratio , etc.). The wind product called L2B contains the final horizontal projection of the LOS wind speed profiles of the Rayleigh and Mie channels, where all necessary calibration and instrument corrections have been performed (Dabas et al., 2008). This product is suitable for the assimilation in NWP models and scientific research. The L2B product also provides scene classification based upon the backscatter ratio corresponding to the wind originating from a 'cloudy' or 'clear' atmospheric region, resulting in Rayleigh-clear, Rayleigh-cloudy, Mie-clear and Mie-cloudy observation types. Throughout the processing chain, the L1B and L2B processors are continuously updated into different baseline versions to account for revisions and identified problems. This leads to different HLOS wind observations and quality in different time periods.

In this study, data from the near-real-time version Baseline product 12 (L2bP 3.50) are used. We evaluate all observation types and corresponding Error Estimates (EEs) of the L2B product except Mie-clear observations as the Mie signal should is usually weak in clear sky conditions. Additionally, two L1B products are used, namely the scattering ratio (SR) and the useful signal. The SR represents the ratio between the total (molecular and particulate) and the molecular backscatter coefficients. It is strictly equal to or greater than one and describes the contribution of the particles to the backscattered signal. Note that the SRs of the L2B products are not used, as some SRs were manually set to one during the processor baseline to eliminate a cross-talk correction, which had detrimental effects on the wind quality. The useful signal represents the returned signal levels per observation and comprises corrections for the solar background, the dark current and the detection chain offset (DCO). We apply an additional range correction and signal normalization that takes into account the different range bin thickness and





distances between the instruments and the height bins. Due to the sequential implementation of the Fizeau and the Fabry-Perot interometers, signal from Mie scattering can leak into the Rayleigh channel signal. This optical "cross-talk" can cause biases, especially in the case of strong Mie returns, as the Rayleigh-channel assumes pure molecular signal in the processing chain.

Along with many other NWP centers, the data were assimilated in the European Centre for Medium-Range Weather Forecasts (ECMWF) Integrated Forecasting System (IFS) by means of the operational four-dimensional ensemble-variational (4D-EnVar) data assimilation scheme (4D-EnVar). At the end of each assimilation cycle, the feedback files with the Aeolus winds and their model equivalents can be retrieved from the Meteorological Archival and Retrieval System (MARS). These reports contain information on the assimilated observations, their model background (short-range forecast) and analysis equivalents as well as various quality control flags.. In this study, background equivalents of Aeolus observations are used as an additional

reference to validate Aeolus HLOS winds. Note that only Rayleigh-clear and the Mie-cloudy winds are in operational use for NWP.

## 2.2 Radiosondes

**Table 1.** Overview of Aeolus overflights and associated radiosonde profiles.

|  | Week day | Start and stop time | Orbit node | Co-location radius | Number of profiles |
|---|---|---|---|---|---|
| **Sal** | Tuesday | 07:28 – 07:29 UTC | Descending | 50 km | 3 |
|  | Thursday | 19:23 – 19:24 UTC | Ascending | 180 km | 3 |
|  | Friday | 19:36 – 19:37 UTC | Ascending | 280 km | 3 |
| **Saint Croix** | Monday | 10:17 - 10:18 UTC | Descending | 90 km | 3 |
|  | Wednesday | 22:12 – 22:13 UTC | Ascending | 160 km | 3 |
|  | Thursday | 22:25 – 22:26 UTC | Ascending | 340 km | 1 |
| **Puerto Rico** | Tuesday | 10:29 – 10:30 UTC | Descending | 160 km | 2 |
|  | Thursday | 22:25 – 22:26 UTC | Ascending | 100 km | 2 |

During the campaign, radiosondes were launched from three different locations over the tropical Atlantic and coordinated by different research components of JATAC. Between the 7 and 28th of September 2021, a total of 37 radiosondes were launched

from Sal airport in Cape Verde, 9 of them corresponding to Aeolus overflights. The launches were coordinated by the Karlsruhe Institute of Technology (KIT) with local support from the JATAC team. This was accomplished using the DFM-09 (GRAW) light weather radiosondes, which measure air pressure, air temperature, relative humidity, wind speed and wind direction. The vertical resolution depends on the ascent speed, which varies with the amount of helium in the balloon, but can generally be estimated at about 5 $ms^{-1}$. Most of the radiosondes launched at Sal were ingested into the Global Telecommunication System

(GTS).



The radiosondes launched on the Virgin Islands were organised by National Aeronautics and Space Administration (NASA)'s Convective Processes Experiment-Aerosols and Winds (CPEX-AW) campaign component of JATAC, with the University of Utah conducting the launches on Saint Croix and the University of Oklahoma conducting the launches from Puerto Rico. On Saint Croix, launches were conducted from Carambola between 19 August 2021 and 14 September 2021. Altogether 73
launches were conducted, of which a total of seven radiosondes were used to validate Aeolus in this study. As for Sal, these measurements were performed with the radiosonde instrument DFM-09 (GRAW). Lastly, 32 launches were conducted from the University of Puerto Rico at Mayagüez (UPRM) campus between 26 August and 14 September 2021, 7 of which could be used for the validation of Aeolus. All launches were performed with iMet-4 radiosondes from the International Met System. As with DFM-09, the iMet-4 radiosondes provide measurements of wind speed, wind direction, temperature, humidity and
air pressure. The radiosonde data also underwent a quality control check using the Atmospheric Sounding Processing Environment (ASPEN) software (Martin and Suhr, 2021) developed by the Earth Observing Laboratory at the National Center for Atmospheric Research (NCAR). A summary of the radiosonde launches and weather events sampled at UPRM was provided by Rios-Berrios et al. (2023).

The total number of radiosonde profiles corresponding to Aeolus overpasses thus amounts to 20, of which 12 correspond to
ascending and 8 to descending orbits of Aeolus. An overview of the launches from the different sites can be found in Table 1, along with other co-location parameters fully discussed in Section 3.1.

## 2.3  EUMETSAT SAFNWC Cloud type product

The Satellite Application Facility for supporting NoWCasting and very short range forecasting (SAFNWC; Alonso Lasheras et al., 2005) developed a number of satellite-based meteorological products distributed by the European Organisation for the
Exploitation of Meteorological Satellites (EUMETSAT). Among others, they provide the Cloud Type (CT) product (Derrien and Le Gléau, 2005), which is a detailed scenery classification of clouds based on different main classes.

The baseline data originate from the Spinning Enhanced Visible and Infrared Imager (SEVIRI) operated onboard the second generation METEOSAT geostationary satellites (MSG). Multispectral thresholding techniques (Saunders and Kriebel, 1988; Derrien et al., 1993; Stowe et al., 1999) are subsequently applied in the NWCSAF software to process the SEVIRI/MSG images
into the various NWC products. The product is available with a temporal resolution of 15 minutes and a nadir spatial resolution of 3 km, compared to 11 km at the edge of the field of view.

In this study, CT is used to identify the cloud type and cloud cover along the Aeolus tracks and to assess the quality of the Aeolus wind products relative to the presence of clouds. More specifically, we identify the pixels closest to each track of Aeolus and determine the average percentage of cloud cover at each altitude based on a cloud classification. According to this
classification, a measurement bin is considered as cloudy, if it is situated within or below a cloud. This refers to following classes for altitudes above 16 km (very high clouds), between 7 and 16 km (very high and high cloud types), between 3 and 7 km (very high, high, mid-level, low and fractional cloud types) and finally below 3 km (very high, high, mid-level, low, very low and fractional cloud types).



## 2.4 CAMS Dust products

The fourth generation of ECMWF Global Atmospheric Composition Reanalysis (EAC4) (Inness et al., 2019) is produced by the Copernicus Atmosphere Monitoring Service (CAMS) with the main objective of global aerosol monitoring. EAC4 relies on ECMWF's IFS, which has been extended to predict and assimilate aerosols (Rémy et al., 2019), trace gases (Flemming et al., 2015; Huijnen et al., 2019) and greenhouse gases. The IFS meteorological and atmospheric composition models are combined with data assimilation from satellite products using the 4D-Var data assimilation scheme in CY42R1. In particular, CAMS

assimilates the Aerosol Optical Depth (AOD) at 550 nm derived from MODIS and the Polar Multi-Sensor Aerosol Optical Properties (PMAp). Reanalysis outputs are provided on three-dimensional time-consistent fields interpolated on 25 pressure levels, a horizontal resolution of about 80 km and a sub-daily time resolution of 6 hours.

Similar to the SAFNWC CT, the dust-aerosol mixing ratio is used to assess the quality of the Aeolus wind products in presence of dust. The dust-aerosol mixing ratio is thereby averaged along each track and projected onto Rayleigh-clear and

Mie-cloudy measurement bins to obtain an estimate of the dust concentration for each observation.

## 3 Methods

### 3.1 Co-location criteria

For the comparison of Aeolus against radiosonde profiles, several steps are required to fit the radiosonde wind measurements to the Aeolus measurement grid and to co-locate them in time and space.

To ensure vertical consistency, the high-resolution radiosonde measurements are vertically averaged within the 24 range bins as specified in the Aeolus L2B product. Subsequently, the radiosondes total horizontal wind speed $V_{RS}$ and direction $\varphi_{RS}$ are projected to the Aeolus HLOS ($HLOS_{RS}$) using the azimuth angle $\varphi_{AEOLUS}$ also specified in the L2B product, in accordance to

$$HLOS_{RS} = V_{RS} \times \cos(\varphi_{AEOLUS} - \varphi_{RS}) \tag{1}$$

Moreover, we have chosen co-location radii of up to 340 km, as we assume typical variations in zonal wind to be of a larger scale. In fact, during boreal summer, African Easterly Waves (AEWs) and tropical disturbances dominate the tropospheric zonal wind variability over the tropical Atlantic, which generally have a horizontal wavelength of 2000-5000 km with a periodicity of 2-7 days (Belanger et al., 2016). Section 4.3.2 discusses the error dependencies related to co-location aspects in more detail.

### 3.2 Statistical metrics

Different metrics were used to validate and estimate the systematic and random error of Aeolus wind products. The bin-to-bin wind speed difference between Aeolus and radiosonde along the HLOS is defined as

$$\Delta_{diff_{HLOS}} = (HLOS_{AEOLUS} - HLOS_{RS}) \tag{2}$$





Thus, the bias $\mu$ is defined as the total mean difference

$$\mu = \frac{1}{N} \sum_{i=1}^{N} \Delta_{\text{diff}_{\text{HLOS}}} \tag{3}$$

with the Mean Absolute Difference (MADI) yielding

$$\text{MADI} = \frac{1}{N} \sum_{i=1}^{N} |\Delta_{\text{diff}_{\text{HLOS}}}| \tag{4}$$

and N the total number of data points.

Additionally, we calculated the standard deviation of the difference

$$\text{STD} = \sqrt{\frac{1}{i-1} \sum_{i=1}^{N} (\text{HLOS}_{\text{AEOLUS}} - \text{HLOS}_{\text{RS}})^2} \tag{5}$$

and the scaled median absolute deviation (SMAD)

$$\text{SMAD} = 1.4826 \times \text{median} \left( |\Delta_{\text{diff}_{\text{HLOS}}} - \text{median}(\Delta_{\text{diff}_{\text{HLOS}}})| \right) \tag{6}$$

The SMAD is equivalent to the standard deviation for a normal distribution of errors, but is often used in Aeolus validation studies as it is less sensitive to individual outliers with very large differences than the standard deviation.

Since the number of data points varies greatly depending on the measurement channel and height, we define the uncertainty

of the mean bias $\varepsilon_\mu$ as

$$\varepsilon_\mu = \frac{\text{SMAD}}{\sqrt{N}} \tag{7}$$

### 3.3 Representativeness

The difference between Aeolus and radiosonde observations is the sum of the Aeolus observation error, the radiosonde observation error and the error arising from spatial and temporal displacement of the observations and different observation

geometries. The latter is usually referred to as representativeness error (Weissmann et al., 2005). As the three error components can be assumed to be uncorrelated, the standard deviation of the Aeolus HLOS winds observation error ($\sigma_{\text{Aeolus}}$) can therefore be calculated as

$$\sigma_{\text{Aeolus}} = \sqrt{\sigma_{\text{tot}}^2 - \sigma_{\text{RS}}^2 - \sigma_{\text{rep}}^2} \tag{8}$$

where $\sigma_{\text{tot}}$ is the standard deviation of the total difference between Aeolus and radiosonde observations (STD), $\sigma_{\text{RS}}$ is the

standard deviation of the radiosonde observation error and $\sigma_{\text{rep}}$ is the standard deviation of the representativeness error. Martin et al. (2021) estimated that the representativeness error for the comparison of Aeolus and radiosonde observations in mit-latitudes is about 2.5 ms$^{-1}$ based on high-resolution model simulations. As the wind fields in the area of the present validation



study is comparably homogeneous, we estimate the representativeness error for our comparison to be in the range of 1.5 ms$^{-1}$ to 2.5 ms$^{-1}$. The radiosonde observation error $\sigma_{RS}$ is estimated to be 0.7 ms$^{-1}$ based on Dirksen et al. (2014).

The representativeness and radiosonde observations errors also need to be considered when comparing the differences between Aeolus and radiosonde observations with the expected error provided in the Aeolus data product (EE$_{Aeolus}$). To account for this, we add the the radiosonde observation error and an estimated representativeness error of 2 ms$^{-1}$ to achieve the total expected error for the comparison (EE$_{tot}$) as follows:

$$EE_{tot} = \sqrt{EE_{Aeolus}^2 + \sigma_{RS}^2 + \sigma_{rep}^2} \tag{9}$$

## 3.4 Quality control

Quality control (QC) is an important step in the evaluation of Aeolus wind errors. The aim is to check for the validity of the measurements and discard nonphysical wind results from the analysis process. The QC we apply here is based on the existing quality control recommendations (Rennie et al., 2020) from the Aeolus Data Science and Innovation Cluster (DISC), and primarily rely on the HLOS wind error estimate (EE) in the L2B product and the validity flags.

The Rayleigh channel EE is based on the uncertainty of the SNR spectrometer response and takes into account error propagation arising from the sensitivity of the Fabry-Perot interferometer, Poisson noise in the useful signal and the solar background. Ultimately, the Rayleigh EE is proportional to the inverse squared root of the useful signal on the detector. Future baseline versions will include additional noise terms, such as noise related to atmospheric temperature and pressure, or cross-talk contamination. In contrary, the Mie EE is determined from the accuracy of the fringe peak position using the solution covariance

of the Lorentzian fitting algorithm based on four characteristics of the signal shape, i.e the peak position, height, width and offset.

Following the default QC flags, all Aeolus wind products with a validity flag of 0, EE above 8 ms$^{-1}$ for Rayleigh and 4 ms$^{-1}$ for Mie, are omitted. Nevertheless, the used QC might not be enough and the data algorithm may contain gross errors in the wind estimate that have not been flagged as invalid. These errors are usually due to non-Gaussian error sources, such as

instrument/transmission failure, or to a misrepresentation of the measurements in space and time. Since the two aforementioned QC are not sufficient to remove these gross errors, an additional QC parameter is used, namely the modified Z-score (Lux et al., 2022b; Witschas et al., 2022; Iglewicz and Hoaglin, 1993). The modified Z-score $Z_{m,i}$ is defined as

$$Z_{m,i} = \frac{\Delta_{diff_{HLOS}} - median(\Delta_{diff_{HLOS}})}{SMAD} \tag{10}$$

and describes the median deviations between each wind speed difference normalized with the SMAD. The modified Z-score

significantly influences small data sets, such as those used in this study. Following literature recommendations (Lux et al., 2022b; Witschas et al., 2022; Sandbhor and Chaphalkar, 2019; Tripathy et al., 2013), we discard wind observations with a modified Z-score greater than 3 as a final QC.




# 4 Results

## 4.1 Statistical comparison of Aeolus with radiosonde observations and model winds

In this section, the L2B HLOS winds (L2bP 3.50) from Aeolus are compared statistically with radiosonde observations and model winds. This includes a comparison with the ECMWF model equivalents (subsection 4.1.1), an overview of systematic and random differences with respect to Cal/Val sites and orbital nodes (subsection 4.1.2), and finally the identification of an orbital- and altitude-dependent bias in the Rayleigh-clear channel (subsection 4.1.3).

The present study relies on a total of 384 Rayleigh-clear and 59 Mie-cloudy bin pairs, of which $\sim 60\%$ and $\sim 53\%$ are
from ascending orbits, respectively, with the majority of observations obtained from the Caribbean launch sites ($\sim 56\%$ for Rayleigh-clear and $\sim 64\%$ for Mie-cloudy). Rayleigh-cloudy bin pairs are also available, but only in a very small number (16 counts), which makes a statistical analysis difficult.

### 4.1.1 Comparative analysis with the ECMWF model equivalents

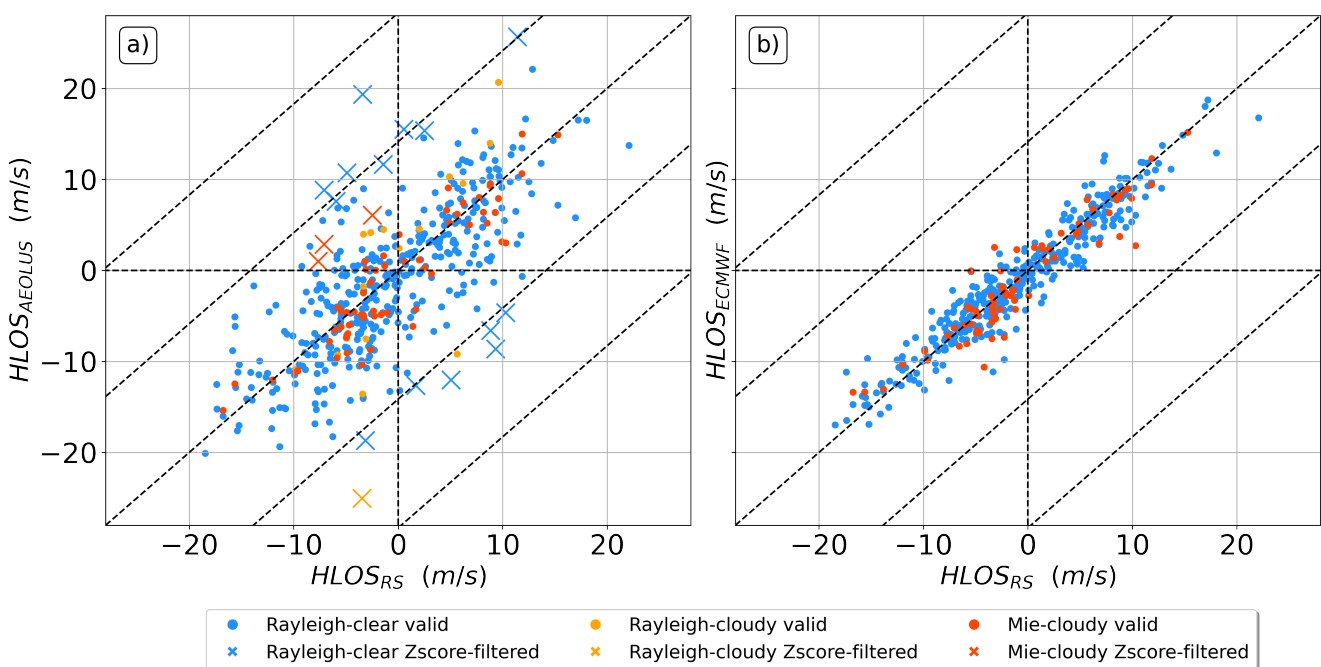

**Figure 1.** (a) Aeolus HLOS Rayleigh-clear (blue), Mie-cloudy (red) and Rayleigh-cloudy (orange) wind products plotted against radiosonde measurements projected along the HLOS for the 20 radiosonde profiles. The gross errors (crosses) are determined using the modified Z-score with a threshold of 3. (b) Aeolus HLOS model equivalents from the ECMWF feedback files plotted against radiosonde measurements. The dashed lines are located at the $\pm 10\,\mathrm{ms}^{-1}$ and $\pm 20\,\mathrm{ms}^{-1}$ wind speed difference between two measurements.



Figure 1 shows a scatter plot of the radiosonde HLOS (HLOS$_{RS}$) against Aeolus L2B (HLOS$_{AEOLUS}$) Rayleigh-clear (blue),
Mie-cloudy (red) and Rayleigh-cloudy (orange) wind products (a) as well as against Aeolus ECMWF model equivalents
(HLOS$_{ECMWF}$) (b). Since Rayleigh-cloudy wind observations are not assimilated at ECMWF, they are not displayed in Fig.
1b. The × symbol represent the gross errors rejected with a Z-score threshold of 3 ($\sim$ 3.5%, $\sim$ 4.8% and $\sim$ 6.7% of the
total Rayleigh-clear, Mie-cloudy and Rayleigh-cloudy data points, respectively). The dashed lines represent the $\pm 10$ ms$^{-1}$
and $\pm 20$ ms$^{-1}$ difference between two measurements. The Aeolus model equivalent HLOS$_{ECMWF}$ for Rayleigh-clear shows a
much better agreement with the radiosonde measurements HLOS$_{RS}$ with a STD of 2.1 ms$^{-1}$ (Fig. 1b) compared to the Aeolus
HLOS$_{AEOLUS}$ Rayleigh-clear observations, which have a larger spread and a STD of 4.8 ms$^{-1}$ (Fig. 1a). The systematic
difference of the model equivalent is also smaller with a bias of $0.1 \pm 0.1$ ms$^{-1}$ compared to $-0.5 \pm 0.2$ ms$^{-1}$ for the Aeolus
observations. In contrast, the Mie-cloudy winds of both Aeolus model equivalents and HLOS$_{AEOLUS}$ behave similarly with
respect to the radiosonde measurements, with STD of 2.93 ms$^{-1}$ and 2.9 ms$^{-1}$, respectively. Again, the systematic difference
in the model equivalent is smaller than for Aeolus Mie-cloudy winds, with biases of $0.4 \pm 0.3$ ms$^{-1}$ and $-0.9 \pm 0.3$ ms$^{-1}$,
respectively. For Rayleigh-cloudy, the STD is larger at 6.6 ms$^{-1}$ with a bias of $1.0 \pm 1.4$ ms$^{-1}$, but given the small statistical
sample size, there is a risk of a large margin of error. The generally good agreement between radiosonde and model equivalent
shows that the co-location parameters used in this study are reliable, as most of the systematic and random errors seem to be
specific to the Aeolus Rayleigh-clear data. This stresses the need to identify the underlying potential error sources of Rayleigh-
clear observations with respect to the presence of clouds and dust aerosols, which are frequent in the region of interest. It is
also worth noting that this good agreement indicates that the model equivalent is a robust reference for validating the Aeolus
winds in the tropical Atlantic.

### 4.1.2 Systematic and random errors using radiosondes

An overview of the bias and random differences of both channels can be found in Table 2. In terms of systematic errors,
Rayleigh-clear shows a relatively small negative bias of $-0.5 \pm 0.2$ ms$^{-1}$, on average, which is below ESA's specification of
0.7 ms$^{-1}$ (Ingmann and Straume, 2016). This bias is, however, the result of a large heterogeneity with respect to the Cal/Val
sites and orbital nodes, with compensating biases of $-1.5 \pm 0.6$ ms$^{-1}$ and $0.6 \pm 0.4$ ms$^{-1}$ for the descending and ascending nodes
on Sal, respectively, compared to negative biases of $-1.0 \pm 0.3$ ms$^{-1}$ (ascending) and $-0.6 \pm 0.4$ ms$^{-1}$ (descending) in the Virgin
Island. As for random differences, Rayleigh-clear has an average STD of 4.8 ms$^{-1}$, which varies only marginally between the
Cal/Val sites and orbital nodes, ranging from 4.1 ms$^{-1}$ to 5.3 ms$^{-1}$. The overall SMAD is found to be slightly below at 4.3 ms$^{-1}$.

For comparison with the ESA recommendation for random errors, we derived the random errors for Aeolus observations
considering also the representativeness errors for the comparison and radiosonde observation errors according to Eq. 8 (table
3). The random error at 2–16 km altitude of 3.8 – 4.3 ms$^{-1}$ exceeds the threshold of 2.5 ms$^{-1}$, while at 16–20 km altitude it
amounts to 4.3 – 4.8 ms$^{-1}$, also exceeding the ESA threshold of 3 ms$^{-1}$. The quality of Rayleigh-clear measurements primarily
depends on the signal accumulation, which can vary with the thickness of the RBS and the horizontal accumulation length
as well as with the atmospheric path signal. The latter has been decreasing in recent years as a result of initial instrumental



**Table 2.** Overview of the mean bias and uncertainty ($\mu$, $\sigma_\mu$; ms$^{-1}$), Standard deviation (STD; ms$^{-1}$), Scaled Median Absolute Deviation (SMAD; ms$^{-1}$) and counts (COUNT) for the Rayleigh-clear and Mie-cloudy channels, orbital nodes and the different radiosonde locations. Due to the small amount of available data, Rayleigh-cloudy is not shown here.

| Region | Orbital node | Rayleigh-clear | | | | Mie-cloudy | | | |
| --- | --- | --- | --- | --- | --- | --- | --- | --- | --- |
| | | $\mu$ | STD | SMAD | COUNT | $\mu$ | STD | SMAD | COUNT |
| | Ascending | 0.6±0.4 | 4.9 | 4.4 | 112 | -1±0.9 | 2.9 | 3.5 | 15 |
| Sal | Descending | -1.5±0.6 | 4.6 | 4.8 | 55 | -1.6±0.8 | 2.2 | 2.1 | 6 |
| | All | -0.1±0.3 | 4.9 | 4.5 | 167 | -1.2±0.7 | 2.7 | 3.2 | 21 |
| | Ascending | -1.0±0.3 | 4.1 | 3.7 | 119 | -0.6±0.7 | 2.9 | 3.7 | 16 |
| SCRX/PR | Descending | -0.6±0.4 | 5.3 | 4.3 | 98 | -1.0±0.5 | 2.9 | 2.5 | 22 |
| | All | -0.8±0.3 | 4.7 | 4.3 | 217 | -0.8±0.4 | 2.9 | 2.5 | 38 |
| | Ascending | -0.2±0.3 | 4.6 | 4.2 | 231 | -0.8±0.6 | 2.9 | 3.3 | 31 |
| Sal/SCRX/PR | Descending | -0.9±0.4 | 5.0 | 4.6 | 153 | -1.1±0.4 | 2.8 | 2.2 | 28 |
| | All | -0.5±0.2 | 4.8 | 4.3 | 384 | -0.9±0.3 | 2.9 | 2.6 | 59 |

misalignment, laser-induced contamination, as well as the wavefront error of the 1.5 m telescope. The solar background noise, which varies along the orbit and season, can also affect the quality of the Rayleigh-clear measurements.

305 For Mie-cloudy, the systematic difference indicates a bias of $-0.9 \pm 0.3$ ms$^{-1}$, which is within the uncertainty range of the ESA's specification and more uniform across regions and orbital nodes with a slightly larger bias in the descending orbits and over Sal. Concerning the random differences, the measurements exhibit a total random error of $1.1 - 2.3$ ms$^{-1}$, which is below ESA's 2–16 km recommendation, as most Mie-cloudy measurements are located underneath 16 km altitude. As with the bias, the STD and SMAD of Mie-cloudy are also quite independent of orbital and regional dependence. The overall accuracy of Mie-cloudy depends on the signal accumulation, the classification algorithm and the quality of the calibration data. The

310 accuracy of Mie-cloudy winds is higher than that of Rayleigh-clear winds as particle backscatter is usually stronger than that of clear air in addition to the fact that Mie backscatter is not subject to broadening induced by Rayleigh-Brillouin scattering (Witschas et al., 2012).

Comparing the results of different Cal/Val studies is tricky as the influence of geographical regions, atmospheric conditions, decreasing laser energy, product baseline and quality control procedures on the result can be significant and must be considered.

315 In this analysis, comparisons are only made with statistics derived from AVATAR-T airborne-based measurements (Witschas et al., 2022; Lux et al., 2022b), as these were carried out in the framework of the same JATAC campaign. The statistical analysis of AVATAR-T shows systematic errors of $-0.1 \pm 0.3$ ms$^{-1}$ for Rayleigh-clear and $-0.7 \pm 0.2$ ms$^{-1}$ for Mie-cloudy, which are slightly smaller than for radiosondes. However, the random error of $7.1 \pm 0.3$ ms$^{-1}$ for Rayleigh-clear is significantly higher.





**Table 3.** Overview of the total systematic ($\mu$, $\sigma_\mu$; ms$^{-1}$) and random ($\sigma_{Aeolus}$; ms$^{-1}$) errors derived according to Eq. 8 for Rayleigh-clear and Mie-cloudy winds for altitudes ranges 2–16km and 16–20km, as well as the corresponding ESA's error recommendations. The random error $\sigma_{Aeolus}$ was computed for a representativeness error $\sigma_{rep}$ ranging from 1.5 ms$^{-1}$ to 2.5 ms$^{-1}$. For Mie-cloudy, only the altitude range 2–16km is shown for the random error, as Mie-cloudy does not sample sufficiently above 16km.

|  | Rayleigh-clear | | | Mie-cloudy | |
| --- | --- | --- | --- | --- | --- |
|  | $\sigma_{Aeolus}$ 2–16km | $\sigma_{Aeolus}$ 16–20km | $\mu$ | $\sigma_{Aeolus}$ 2–16km | $\mu$ |
| Ascending | $3.4 - 3.9$ | $4.0 - 4.4$ | -0.2±0.3 | $1.1 - 2.3$ | -0.8±0.6 |
| Descending | $4.3 - 4.7$ | $4.4 - 4.9$ | -0.9±0.4 | $0.5 - 2.1$ | -1.1±0.4 |
| All | $3.8 - 4.3$ | $4.3 - 4.8$ | -0.5±0.2 | $1.1 - 2.3$ | -0.9±0.3 |
| ESA | 2.5 | 3 | 0.7 | 2.5 | 0.7 |

The difference in results is caused by the different altitudes at which the data are sampled, as the aircraft only samples the lower 10 km portion of the troposphere, which is shown to be more noisy owing to the abundance of dust aerosols in this region. For Mie-cloudy, the random error gives $2.9 \pm 0.3$ ms$^{-1}$, which is similar to our radiosonde-based results as most Mie-cloudy scattering occurs at lower levels.

### 4.1.3 Orbital bias in the Rayleigh-clear channel

Figure 2 shows vertical profiles of the differences between Aeolus Rayleigh-clear observations and radiosonde measurements projected along HLOS (O-RS; solid lines), and the corresponding ECMWF model equivalents (O-B; dotted lines) for both ascending (red) and descending (blue) orbits over Sal (a), PR and SCRX (b). The shading represents the bias uncertainty $\sigma_\mu$. HLOS winds from the descending track are multiplied by -1 to conform with the sign convention of the model coordinate system. The vertical profiles illustrate the presence of an ascending/descending bias visible in both the O-B and O-RS profiles, reaching up to 2.5 ms$^{-1}$ around 8 km altitude in both regions. The differences below 5 km altitude could be related to the greater amount of dust in Cabo Verde during this period, while above 17 km the differences could partly be related to the lack of descending orbit data over Sal (Fig. 2a). This altitude- and orbit-dependent bias was already described by Borne et al. (2023) using first-guess departure statistics over West Africa.

This latitude consistent bias caused the zonal winds in the ECMWF analysis to accelerate in the morning and weaken in the evening, affecting the African Easterly Jet (AEJ) and Tropical Easterly Jet (TEJ) in particular. Correcting this bias with a temperature-dependent approach helped to improve the representation of winds in the analysis and forecast fields (Borne et al., 2023). However, the cause of this bias remains unknown, as it has not been proven to be related to temperature, nor has any dependence on wind speed, SNR or useful signal been found (not shown here). Here, as both the O-B and O-RS profiles are very close to each other, with deviations below 0.5 ms$^{-1}$, the existence of this bias can be confirmed observationally with radiosondes. As highlighted by Horányi et al. (2015), biases of the order of 1 ms$^{-1}$ can already deteriorate forecast quality.





**Figure 2.** Differences (dots) and average differences (lines) between ascending (red) and descending (blue) winds between Aeolus observations (O) and radiosonde HLOS wind measurements (RS, solid line) along with ECMWF model equivalents (B, dotted line) over Sal (a) and Puerto Rico - Saint Croix (PR/SCRX, (b). The shadow represents the bias uncertainty $\sigma_\mu$. To comply with the sign convention of the model coordinate system, the HLOS winds from the descending orbit are multiplied by -1.

## 4.2 Error dependency

In this section we examine the error dependency and associated error sources of the different Aeolus wind products. Firstly, we investigate the error dependency as a function of co-location parameters, such as radius and time difference between two measurement points, to account for representativeness. Secondly, we explore the error dependency in relation to the presence of clouds and dust, as these supposedly influence the quality of Aeolus wind products.




### 4.2.1 Temporal and spatial co-location

**Rayleigh-clear and Rayleigh-cloudy**

Figure 3 shows the absolute difference between Aeolus and radiosonde measurement points $|\Delta_{\text{diff}_{\text{HLOS}}}|$ as a function of $EE_{\text{tot}}$ (a), altitude (b), co-location radius (c) and co-location time (d) for the Rayleigh-clear (blue) and Rayleigh-cloudy (orange) observation types. The solid and dashed blue lines show the Rayleigh-clear MADI and SMAD, respectively, with each value calculated using a minimum sample size of 40 data points for panels a, b and d. Also shown are outliers (cross symbol +), that we define in this study as values with low EE ($< 5$ ms$^{-1}$) and large absolute difference ($> 10$ ms$^{-1}$), which are of particular interest as they contribute the most to the wind quality degradation. The Rayleigh-clear outliers account for 13 observations, i.e. $\sim 3.4\%$ of the data points. For Rayleigh-cloudy, no MADI and SMAD are computed due to the lack of data.

In general, the MADI and SMAD between Rayleigh-clear and radiosonde wind measurements appear to be proportional to the Aeolus $EE_{\text{tot}}$ (Fig. 3a), with larger deviations associated with larger $EE_{\text{tot}}$s, as expected. However, on average, the mean $EE_{\text{tot}}$ overestimates the MADI by 1 ms$^{-1}$ for $EE_{\text{tot}}$ values below 6 ms$^{-1}$ (see grey line). This discrepancy can be attributed to the relatively small amount of data used in the study, as the EE is based on the Gaussian assumption of a large data set. For Rayleigh-cloudy measurements, it is difficult to establish a dependency although the absolute difference appears to be generally larger owing to the large STD of 6.6 ms$^{-1}$ for this observation type. Considering the altitude error dependency of Rayleigh-clear (Fig. 3b), a general pattern emerges with MADI and SMAD reaching a minimum of 3 ms$^{-1}$ and 2 ms$^{-1}$ respectively on average in the middle troposphere at 10 km, while increasing above and below, with MADIs of 4–5 ms$^{-1}$ and SMADs of almost 6 ms$^{-1}$ at 2.5 km and 19 km altitude. As we will see in the next subsection 4.2.2, this error pattern is inversely proportional to the Rayleigh backscattered useful signal, as it directly affects the SNR and thereby the quality of the measurement points. Rayleigh-clear outliers seem to occur at all altitudes and Rayleigh-cloudy measurements are primarily found in the lower troposphere, below 6 km.

In Fig. 3c we examine the error dependency with respect to the co-location radius, which extends up to 340 km, a distance that is large relative to the 100 km specified in ESA's recommendations. However, the MADI and SMAD for Rayleigh-clear do not increase with radius, but stagnate at an average of 3–4 ms$^{-1}$ for radii above 100 km, while they are slightly higher below 100 km, reaching 4–5 ms$^{-1}$. Furthermore, outliers appear across all co-location radii. This indicates that the use of a co-location distance up to 340 km is acceptable for the statistical comparison. Exploring the error dependency with respect to the time difference between the observations (Fig. 3d), there is indication for increasing difference for larger time-differences, going from 3–4 ms$^{-1}$ at 0 minutes to 4–6 ms$^{-1}$ above 30 minutes. There is also an asymmetry of the error dependence, with a larger error magnitude for radiosonde observations preceding the Aeolus passage. Since most radiosondes were launched with the objective of reaching the mid-troposphere during the satellite's passage, the measurements preceding/following Aeolus of more than 30 minutes correspond mainly to measurements at lower/higher altitudes. The larger MADI and SMADI values for these time differences could hence be an indirect effect of the larger errors found at those altitudes (Fig. 3b). Again, no error dependency is observed for outliers, with most occurring below ±40 minutes time differences.




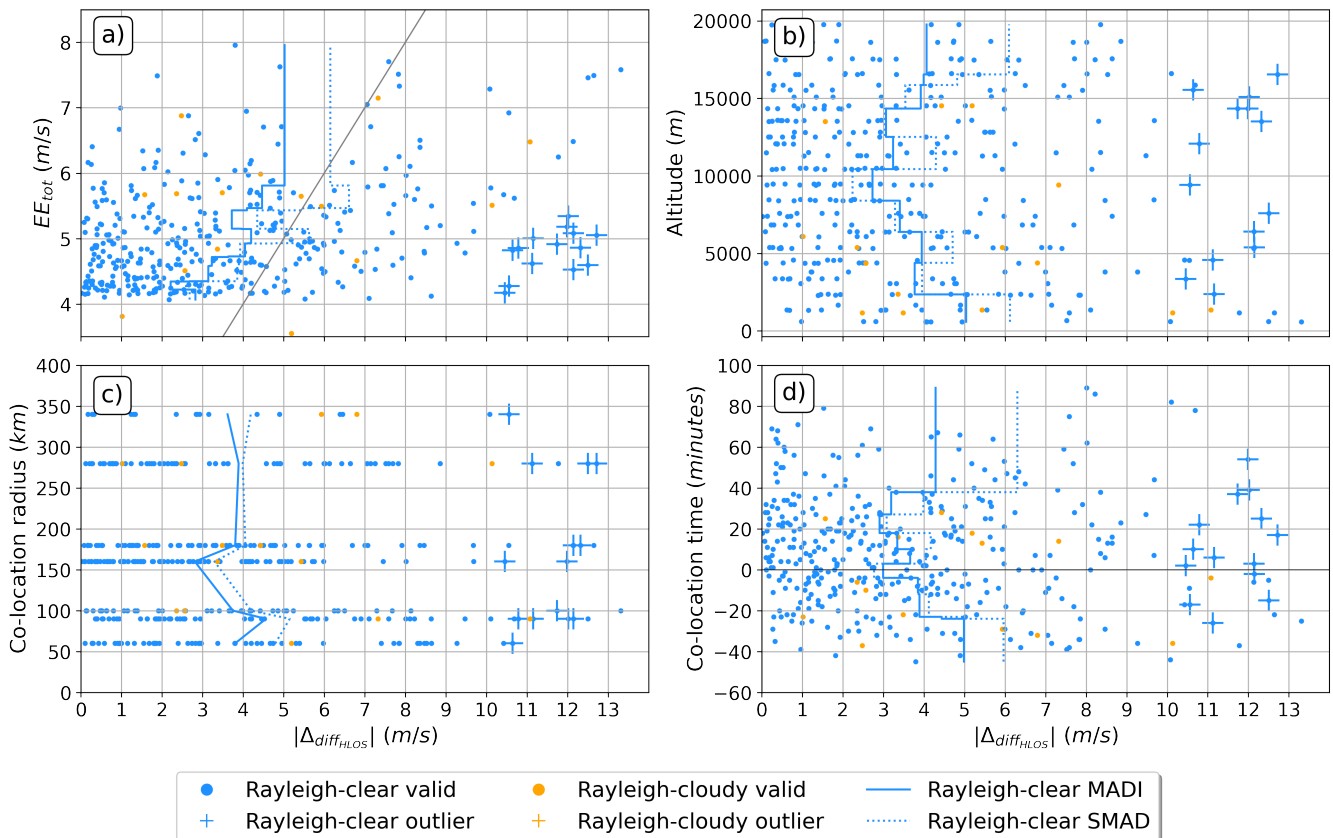

**Figure 3.** $EE_{tot}$ (a), altitude (b), co-location radius (c) and co-location time quantities expressed as a function of the absolute difference between radiosonde HLOS winds ($HLOS_{RS}$) and Aeolus ($HLOS_{AEOLUS}$) Rayleigh-clear (blue) and Rayleigh-cloudy (orange) observations. Outliers are defined as values with an EE below 5 ms$^{-1}$ and absolute difference larger than 10 ms$^{-1}$ and are represented by the cross symbol +. The solid blue lines indicate the MADI while the dotted blue lines represent the SMAD of Rayleigh-clear and each value is computed using a minimum sample size of 40 data points. The grey line in panel a represents the diagonal at intercept 0 with slope 1. Due to the limited amount of data, no MADI and SMAD are shown for Rayleigh-cloudy.

**Mie-cloudy**

Figure 4 shows the same error dependencies as in Fig. 3, but for the Mie-cloudy observation type. For Mie-cloudy, we define
380   outliers as values exceeding an absolute error of 6 ms$^{-1}$ along with EEs inferior to 3 ms$^{-1}$. With a total of 3 data points, they account for $\sim$ 5% of the total Mie-cloudy observations. In panels a, b and d, each MADI and SMAD value is calculated using a minimum sample size of 15 data points

As shown in Fig. 4a, the absolute differences for Mie-cloudy measurements are generally smaller than for Rayleigh-clear, with the largest deviations around 7–8 ms$^{-1}$, while attaining 13–14 ms$^{-1}$ for Rayleigh-clear. The MADI and SMAD remain
385   between 2 and 3 ms$^{-1}$, indicating an overestimation of the $EE_{tot}$, especially for increasing $EE_{tot}$. Regarding the altitude error



dependency (Fig. 4b), most of the data are found within the 10-15 km layer, which is probably related to the presence of high-level clouds, and below 7 km, where low- and mid-level clouds and dust layers are found. Due to the sparseness of Mie-cloudy data, both MADI and SMADI do not show a specific vertical error trend. While MADI and SMAD remain between 2.3 and 2.7 ms$^{-1}$, respectively, they decrease to 1.8 and 2.3 between 1.5 and 3 km altitude before increasing to almost 3 ms$^{-1}$ in the lowest 1 km. Fig. 4c shows that similarly to Rayleigh-clear, Mie-cloudy reveals no error dependency with respect to co-location radii, with the mean absolute error and SMAD mainly ranging from 1.7 to 3.2 ms$^{-1}$, and outliers found at all radii. Regarding the error dependence on time difference (Fig. 4d), we find that most of the measurement differences occur at time intervals of less than ±40 minutes. MADIs and SMADs are generally higher for negative co-location times, corresponding to cases where radiosonde observations are sampled before those from Aeolus. Nevertheless, we do not notice a strong relationship between co-location time and errors.

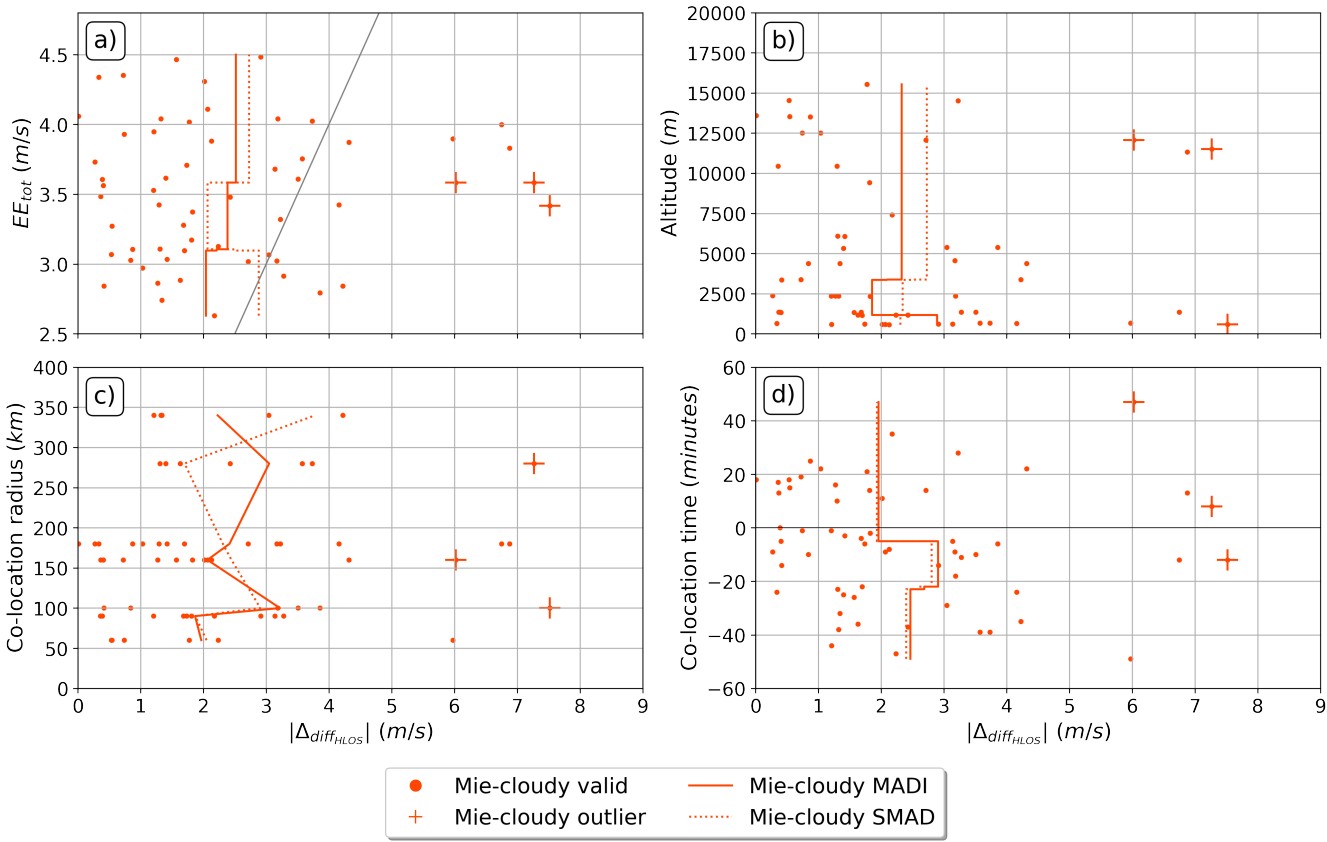

**Figure 4.** Same as for Fig. 3, but for Mie-cloudy. For Mie-cloudy (red), outliers are defined as values having an absolute error above 6 ms$^{-1}$ and an EE inferior to 3 ms$^{-1}$. The MADI and the SMAD values are computed using a minimum sample size of 15 data points.





### 4.2.2 Cloud type and dust

As already mentioned, the accuracy of Rayleigh-clear and, to a lesser extent, Mie-cloudy depends on the signal level and SNR. In general, the signal level depends on the range bin thickness, the horizontal accumulation length, the atmospheric path signal and the overall signal background level. In addition, Rayleigh-clear winds are sensitive to signal attenuation due to atmospheric conditions, with weaker signal return under optically thick clouds and dust-aerosol layers. Mie-cloudy is less concerned as backscatter from particles is stronger, although it is sensitive to weak backscatter, e.g. from dust layers. Because of its strong sensitivity on signal levels, the EE of Rayleigh-clear only considers Poisson noise and is therefore inversely proportional to the squared root of the useful signal. For Mie-cloudy, this rule of thumb is not true. In this context, we aim to investigate the quality of the Rayleigh-clear and Mie-cloudy winds and the reliability of the corresponding EE with respect to the presence of clouds and dust.

**Rayleigh-clear**

Table 4 describes the error dependency of the Rayleigh-clear observations with respect to the presence of clouds and dust, with cases below 50%, above 50% and above 75% of cloudiness, as well as sub-categories distinguishing the dust mixing ratio above (Dust) and below (Dust$_\mathrm{NO}$) $10^{-8}$ kgkg$^{-1}$. Note that SMAD is not used for this analysis as this reliably removes outliers, which ought to be quantified here. We note that the MADI, the STD, and the EE$_\mathrm{tot}$ all increase with the amount of clouds and dust along the track, presumably due to the reduced return signal. In non-dusty conditions (Dust$_\mathrm{NO}$), we observe that for low cloud cover (<50%), the MADI ($3.3 \pm 0.2$ ms$^{-1}$) is significantly lower than the EE$_\mathrm{tot}$ ($4.8$ ms$^{-1}$) with a difference of $1.5$ ms$^{-1}$, while for higher cloud cover, the difference between MADI and EE$_\mathrm{tot}$ is much smaller ($1.1$ ms$^{-1}$ and $1.0$ ms$^{-1}$ for above 50% and 75% of cloudiness, respectively). This phenomenon is further enhanced at higher dust concentrations, with the MADI reaching even higher values ($5.7 \pm 0.8$ ms$^{-1}$) than the EE$_\mathrm{tot}$ ($5.8$ ms$^{-1}$) for cloud cover above 75%. This highlights how the EE$_\mathrm{tot}$ in clear sky conditions is well calibrated, while it is becoming gradually too low with the increasing presence of clouds and dust. The larger STD with increasing cloudiness and dust concentration suggests an increasingly perturbed pattern of Rayleigh-clear measurements, possibly owing to the lower signal levels or to a cross-talk.

Figure 5 puts this phenomenon into perspective, by showing the altitude-dependent absolute difference $|\Delta_{\mathrm{diff_{HLOS}}}|$ (a,e), the EE$_\mathrm{tot}$ (b,f), the normalized useful signal (c,g) and the SR (d,h), where the colouring depends on the percentage of SAF clouds (upper row) and the CAMS dust mixing ratio (lower row) along the track. For reference, the values that did not pass the QC are shown transparently. In addition, panel 5a includes the MADI of four cloud cover percentage categories, where each MADI is computed with a minimum sample size of 10 values. The colourings in Fig. 5 are illustrative of the results summarised in Table 4, with measurements showing generally greater MADI under high cloud cover (red, orange, Fig. 5a) than under lower cloud cover (blue, blue-green). Measurements in the lower troposphere are naturally more strongly affected by cloud cover compared to higher levels. The same applies to dust (Fig. 5e), which also occurs mainly in the lower 5 km of the troposphere.

As we have already shown in Fig. 3b, the absolute error is higher in the upper and lower troposphere and minimised in the middle troposphere around 10 km altitude. This trend is well reflected in the EE$_\mathrm{tot}$ in Fig. 5b, which is an indication of



**Table 4.** Overview of the total Error Estimate (EE$_{tot}$; ms$^{-1}$), mean absolute difference and uncertainty (MADI, $\sigma_\mu$; ms$^{-1}$), Standard deviation (STD; ms$^{-1}$) and counts (COUNT) for the Rayleigh-clear measurements under different cloud and dust conditions. This includes three categories of cloud cover (< 25 %, > 50 %, > 75 %) for dust mixing ratios above (Dust) and below (Dust$_{NO}$) $10^9$ kgkg$^{-1}$ along the track.

|  | Cloud < 50 % | | Cloud > 50 % | | Cloud > 75 % | |
|---|---|---|---|---|---|---|
|  | Dust$_{NO}$ | Dust | Dust$_{NO}$ | Dust | Dust$_{NO}$ | Dust |
| EE$_{tot}$ | 4.8 | 5.4 | 5.0 | 5.6 | 5.3 | 5.8 |
| MADI | 3.3±0.2 | 4.4±0.6 | 3.9±0.5 | 5.0±0.5 | 4.3±0.7 | 5.7±0.8 |
| STD | 4.3 | 5.0 | 5.1 | 5.9 | 5.6 | 6.4 |
| COUNT | 234 | 28 | 64 | 52 | 38 | 24 |

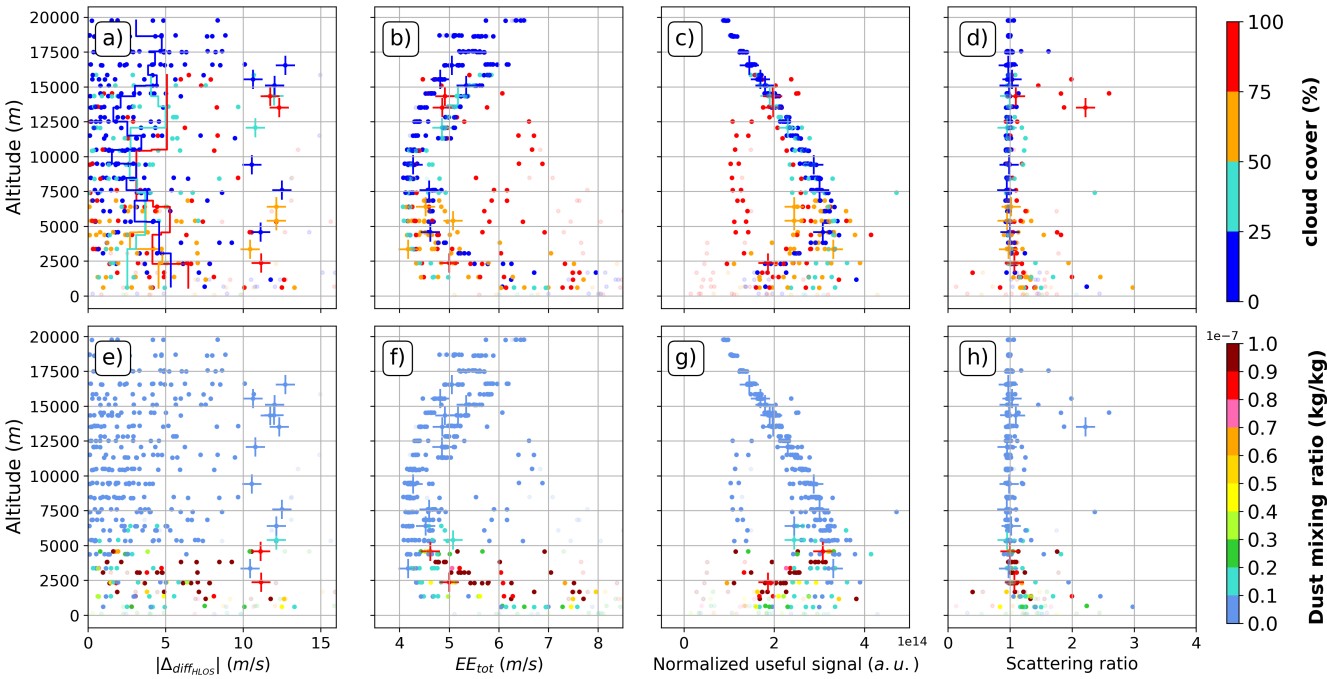

**Figure 5.** Altitude as a function of Rayleigh-clear absolute difference $|\Delta_{\text{diff}_{HLOS}}|$ (a,e), EE$_{tot}$ (b,f), normalized useful signal (c,g) and SR (d,h), where the colouring is dependent on the percentage of SAF clouds (upper row) and CAMS dust mixing ratio (lower row) along the track. The cross symbol + stands for outliers and defines values with an EE below 5 ms$^{-1}$ and an absolute difference of more than 10 ms$^{-1}$. Panel (a) includes the MADI for each cloud cover percentage, with a minimum sample size of 10 data points used to compute each value.

the generally good consistency between the EE$_{tot}$ and the absolute differences. As expected, this tendency fits inversely with
the normalized useful signal shown in Fig. 5c, with lower signal in the upper and lower troposphere. Indeed, in the higher troposphere the air is less dense and the thickness of the RB's is not sufficient to compensate for the decrease in air molecule





density. In the lower troposphere, the return signal is lower due to strong attenuation under clouds and dust layers. Interestingly, the values with high $EE_{tot}$ and smaller useful signal in the mid-troposphere between 5 and 12.5 km in red likely correspond to measurements sampled under thick clouds, resulting in a strongly attenuated signal. They account for most of the measurements

with cloud cover greater than 75 % in this altitude range, while the cloud tops appear to be located between 12.5 and 15 km, as they exhibit a larger normalized useful signal and a SR greater than 1 (Fig. 5d,h). Finally, outliers are found under all types of cloud and dust conditions and affect different altitude ranges. They also occur for regular normalized useful signals, with most SRs lying around 1, which rules out a cause related to atmospheric particles.

**Mie-cloudy**

Table 5 shows the same as Table 4, but for Mie-cloudy. Due to the limited amount of data for Mie-cloudy winds, the interpretation of the results should be treated with caution. We find that, in contrast to Rayleigh-clear, the EE, MADI and STD decrease with the percentage of cloud cover along the path. This is understandable as clouds provide the strongest backscatter signal required for high quality Mie-cloudy measurements. However, the presence of dust for cloud cover below 50 % leads to a decrease in $EE_{tot}$, MADI and STD, while conversely there is an increase of these quantities in more dense cloudy conditions

(>50 %, >75 %). A possible explanation is that in clear-sky conditions, the backscatter from dust layers is strong enough to obtain high quality measurements, whereas in cloudy conditions, the attenuation by clouds weakens the backscatter return from the dust.

Figure 6 depicts the same as Fig. 5, but for Mie-cloudy. As mentioned in the previous section when discussing in Fig. 4b, most backscatter occurs in two layers, i.e. within 10–15 km and below 7 km altitude. The majority of measurements have

normalized useful signals above $5e^{13}$ a.u. (Fig. 6c,g), which is overall above the normalized useful signal of the rejected measurements shown in transparent. Furthermore, the SRs are generally above 1 (Fig. 6d,h), which is characteristic of Mie-cloudy measurements. More specifically, measurements sampled above 12.5 km have a cloud cover of more than 75 % along the track and probably correspond to cloud tops, as they have stronger SRs between 1.5 and 3 (Fig. 6d,h). They exhibit good quality as well, with an average MADI of 1.5 ms$^{-1}$ (Fig. 4b). Between 7.5 and 12.5 km altitude, most of the measurements

occur with cloud cover less than 50 %, with SRs falling below 1.3. In this altitude range, there are also 2 outliers, which interestingly have SRs around 1 and a normalized useful signal in the same order of magnitude as the discarded ones. Their presence is unusual, as Mie-cloudy measurements are only obtainable for SRs above 1. Finally, below 7.5 km, the cloud cover is mainly above 50 %, while the dust concentration is mainly below $5 \times 10^{-8}$ kgkg$^{-1}$, showing that most of the Mie-cloudy backscatter results from clouds and not from dust. As can be seen in Fig. 6g, measurements with high dust concentration

(brown) are discarded (transparent) with normalized useful signals below $5e^{13}$ a.u. Surprisingly, measurements sampled at the lower 1 km have the lowest normalized useful signals, mostly below $5e^{13}$ a.u. and are not discarded. These, however, tend to have larger SRs between 1 and 2, which can compensate for the low normalized useful signal in the calculation of the EE. They also correspond to the largest MADI scaling up to 4 ms$^{-1}$ on average (i.e. Fig. 4d) in addition to relatively high $EE_{tot}$s (i.e. Fig. 6b,f). Two measurements also show negative SRs, which is an artifact, due to insufficient background signal corrections.

The third outlier in the lower 1 km does not have abnormal characteristics compared to other measurements at this altitude.




**Table 5.** Same as table 4, but for Mie-cloudy.

|  | Cloud < 50 % | | Cloud > 50 % | | Cloud > 75 % | |
|---|---|---|---|---|---|---|
|  | Dust$_{NO}$ | Dust | Dust$_{NO}$ | Dust | Dust$_{NO}$ | Dust |
| EE$_{tot}$ | 3.7 | 3.6 | 3.4 | 3.5 | 3.2 | 3.4 |
| MADI | 2.8±0.5 | 2.4±0.2 | 1.8±0.3 | 2.5±0.4 | 1.6±0.3 | 2.6±0.6 |
| STD | 2.96 | 1.53 | 1.89 | 2.95 | 1.68 | 3.18 |
| COUNT | 11 | 9 | 16 | 23 | 8 | 13 |

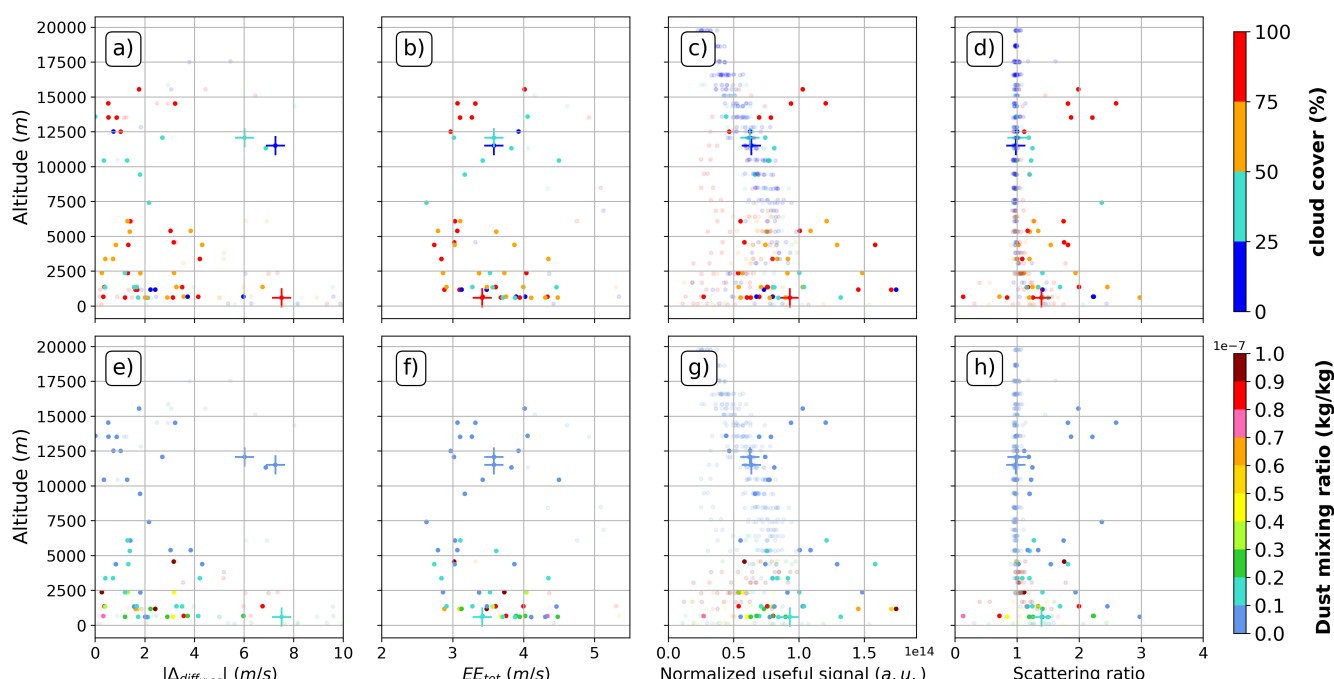

**Figure 6.** Same as Fig. 5 but for Mie-cloudy. Here the cross symbol + defines values with an EE below 3 ms$^{-1}$ and an absolute difference above 6 ms$^{-1}$.

### 4.2.3 Case studies

To further investigate the properties of the Aeolus wind errors, this section presents three case studies comparing Aeolus and radiosonde wind measurements under three different atmospheric conditions, namely clear sky, high cloud cover and high dust concentration.

The first case-study illustrated in Fig. 7 presents a comparison between Aeolus and radiosonde wind measurements collected under clear sky conditions. The radiosonde was launched over Sal Airport at 18:45 UTC on 9 September 2021, and Aeolus



**Figure 7.** Overview of the cloud-free case study for a radiosonde launched from Sal airport on 9th September 2021 at 18:45 UTC and the ascending orbit of Aeolus between 19:23:56-19:24:31 UTC for a co-location radius of 180km. (a) Vertical radiosonde HLOS wind profile (black solid line) and projected onto Rayleigh-clear RBS (black stepped line), as well as averaged Rayleigh-clear observations (blue dots), corresponding EE (error bars) and ECMWF model equivalents ($M_{eq}$, stepped lines). (b) Vertical profile of the Rayleigh-clear $EE_{tot}$ (blue line), together with the $EE_{tot}$ of all profiles (grey solid lines) and their average (black solid line). (c), (d) Same as (b), but for normalised useful signal and CAMS dust mixing ratio, respectively. (e) Horizontal map showing the SAFNWC CT at 19:00 UTC and the co-location perimeter (white solid line), the Aeolus track (red solid line) and the radiosonde launch site (red cross).

passed over on an ascending orbit between 19:23:56 UTC and 19:24:31 UTC within a co-location radius of 180 km around the launch site. Figure 7a depicts the corresponding sampled radiosonde HLOS wind profile (black lines) as well as Rayleigh-clear (blue) wind measurement points with associated $EE_{tot}$ shown as error bars and ECMWF model equivalents shown as stepped lines. The corresponding Rayleigh-clear $EE_{tot}$, normalized useful signal and CAMS dust mixing ratio profiles are shown in blue in Figs. 7b, 7c and 7d, respectively, along with all other profiles in grey and the average of all profiles in black. Figure






7e shows the SAFNWC CT over the Cape Verde region at 19:00 UTC. In the latter panel, it can be seen that conditions were predominantly cloud-free along the Aeolus track (red solid line) and within the co-location radius (white solid line), while some low clouds can be found in the surrounding area. In these clear-sky conditions, it is not surprising to find that most of the

measurements are of the Rayleigh-clear observation type, with no Mie-cloudy and Rayleigh-cloudy measurements (Fig. 7a). Throughout the atmosphere above 2.5 km, the quality of Rayleigh-clear is very good, with most error bars overlapping with radiosonde measurements and ECMWF model equivalents. In general, we found that the $EE_{tot}$ estimate (Fig. 7b) is below average throughout the atmosphere, with a minimum of 3.5 ms$^{-1}$ at 8 km altitude and a maximum above 5 ms$^{-1}$ at 17.5 km and 2.5 km altitude. This is consistent with to a normalized useful signal (Fig. 7c) close to the average, except between 2.5 and

12.5 km, where it is higher, most likely due to the absence of cloud attenuation. In general, $EE_{tot}$ and normalized useful signal decrease below 5 km, which is accompanied by an increase in the dust mixing ratio. This increase reaches 1.2 kgkg$^{-1}$ at about 2 km altitude, below which no measurements are found, presumably filtered out during the QC procedure.

Figure 8 shows the same as Fig. 7, but for cloudy conditions. In this case study, the radiosonde was also launched from Sal airport, this time at 07:00 UTC on 14 September 2021, with a co-location radius of 60 km. Aeolus passed across the

co-location region between 07:28:32 UTC and 07:28:55 UTC, i.e. during the descending node. As can been seen in panel 8e, which corresponds to SAFNWC CT at 07:30 UTC, Aeolus overpasses a variety of high clouds, mainly high semitransparent clouds. These high-clouds appear to be located between 13 km and 16 km altitude, as three Mie-cloudy (red) and two Rayleigh-cloudy (orange) measurements are found in this range, and where the normalized useful signal is found to have a maximum. In this altitude range, all Rayleigh-clear, Rayleigh-cloudy and Mie-cloudy measurements exhibit good quality, with radiosonde

measurements generally within the error bars. Above this cloud cover at 16 km, we only find Rayleigh-clear measurements that also perform well, with an $EE_{tot}$ (Fig. 8b) and normalized useful signal (Fig. 8c) close to average. Beneath the cloud base at 13 km altitude, however, it appears that the Rayleigh-clear measurements follow an irregular pattern, with most of the measurements and error bars not matching the radiosonde observations, reaching deviations higher than 10 ms$^{-1}$. Accordingly, we find that the $EE_{tot}$ (Fig. 8b) is larger in this altitude range mainly varying between 5 and 6 ms$^{-1}$, which also corresponds to a

sharp decrease of the normalized useful signal well below the average (Fig. 8c). Nonetheless, the ECMWF model-equivalents in Fig. 8a remain fairly accurate relative to the radiosonde measurements. This result mirrors the findings presented in the previous section, namely that the Rayleigh-clear $EE_{tot}$ is systematically underestimated when the normalized useful signal is strongly attenuated. It appears that the normalized useful signal further decreases below 2.5 km, presumably as a result of the increasing dust concentration at this height (Fig. 8d), which most likely leads to a QC rejection of the Rayleigh-clear

measurements.

Lastly, Fig. 9 examines the influence of dust on the quality of Aeolus. In this case, the radiosonde was launched on 21 September 2021 at 06:50 UTC for a descending orbit of Aeolus, which passed over a co-location perimeter with a radius of 60 km between 07:28:44 UTC and 07:29:07 UTC. As can be seen in Fig. 9e, the atmospheric conditions in the co-location area were completely cloud free at 07:30, with some low level cloud further south of the island. The radiosonde profile shown in Fig.

9a indicates that Aeolus primarily measured in the Rayleigh channel along this orbital segment. Rayleigh-clear measurements appear to be consistent with radiosonde wind measurements throughout the mid-troposphere between 5 km and 15 km altitude,





**Figure 8.** Same as Fig. 7, but for the case study with high cloud cover. Here, the radiosonde was launched from the Sal airport at 07:00 UTC on the 14th September 2021, while Aeolus overpassed the co-location area, with radius of 60km, on a descending node between 07:28:32 and 07:28:55 UTC. In panel a, the red and orange colours represent the averaged Mie-cloudy and Rayleigh-cloudy observations (points), respectively, with the corresponding EE shown as error bars and ECMWF model equivalents ($M_{eq}$) shown as stepped lines. The SAFNWC CT shown in (e) corresponds to 07:30 UTC.

while outliers with EEs of less than $5\,\mathrm{ms^{-1}}$ (Fig. 9b) can be spotted above 15km and below 5km. This error structure is surprising, as both the normalized useful signal and error estimation curves are similar to the one of the cloud-free case study in Fig. 7b and 7c. However, in panel 9c, we see that the Rayleigh-clear error pattern coincides with a strong peak in dust
mixing ratio, reaching more than $2\,10^{-7}\,\mathrm{kgkg^{-1}}$ around 3.5 km altitude. The presence of dust seems to affect the quality of Rayleigh-clear measurements without influencing the normalized useful signal and thus leading to an underestimation of the EE. Reason could be linked to a cross-talk.



**Figure 9.** Same as Figs. 7 and 8 but for the case study with dust. Here, the radiosonde was launched from the Sal airport at 06:50 UTC on the 21th September 2021, while Aeolus overpassed the co-location area, with radius of 60km, on a descending node between 07:28:44 UTC and 07:29:07 UTC. The SAFNWC CT shown in (e) corresponds to 07:30 UTC.

# 5   Conclusions

In this study, we conducted a cross-Atlantic validation of Aeolus wind observations using radiosondes in the scope of the Joint Aeolus Tropical Atlantic Campaign (JATAC). Of the total 20 radiosonde profiles included in this work, 11 were launched from Puerto Rico and St. Croix in the Caribbean and 9 from Sal Airport on Cape Verde between August and September 2021. The advantage of radiosondes is that they provide good vertical coverage, providing 384 Rayleigh-clear bin-to-bin comparisons from the surface to an altitude of 20 km and 59 Mie-cloudy comparisons, mainly restricted to the presence of clouds and aerosols. After having applied several Quality Control (QC) and adaptation grid procedures, we quantified the quality of





Rayleigh-clear, Mie-cloudy and to a lesser extent Rayleigh-cloudy observation types, with respect to co-location aspects as well as atmospheric conditions such as cloud cover and dust concentration.

According to our statistical analysis, the total systematic error of Rayleigh-clear is $-0.5 \pm 0.2$ ms$^{-1}$, which is in agreement with the ESA recommendation of 0.7 ms$^{-1}$. The random error was calculated from the standard deviation of the difference between radiosonde and Aeolus measurements, accounting for radiosonde observation errors estimated at $0.7 \pm 0.28$ ms$^{-1}$ and

representativeness errors ranging from 1.5 to 2.5 ms$^{-1}$. In the altitude range of 2–16 km and 16–20 km, the random error is $3.8 - 4.3$ ms$^{-1}$ and $4.3 - 4.8$ ms$^{-1}$, respectively, which is above the ESA-specified values of 2.5 ms$^{-1}$ and 3 ms$^{-1}$, respectively. In general, Rayleigh-clear shows no error dependency with respect to co-location radius, even for distances reaching 340 km, whilst being more sensitive to co-location time, especially if the radiosonde measurement is ahead of Aeolus' over-flight time, which presumably corresponds to low altitude measurements. In addition, the systematic and random errors are

height-dependent, with larger errors occurring in the upper troposphere, mainly caused by the reduction in signal return from decreasing air density, and in lower levels, most likely caused by the signal attenuation by clouds and dust. The error estimate likewise follows a similar form to the observed height error dependency, as it is inversely proportional to the squared root of the normalized useful signal. In cases where the normalized useful signal is strongly attenuated by clouds or dust, the error estimate is generally underestimated, with measurements exhibiting non physical features and departures from radiosonde

winds larger than the error estimate. A redefinition of the Rayleigh-clear error estimate could account for this underestimation by including other sources of noise, such as detector noise or readout noise, which increase for reduced signal levels. Furthermore, a cross-talk, i.e. the leakage of the Mie signal into the Rayleigh receiver, could also explain this underestimation, especially in the case of strong Mie returns. However, this supposition was not investigated in the context of this study. Outliers, defined as measurements with small error estimate and large absolute differences, are found under all conditions, i.e. for all

co-location radii, co-location times, altitudes as well as cloud and dust cover. Their origin does not appear to be correlated with low signal levels but seem to be inherent to the statistical nature of the error distribution. Taking other terms into account when defining the error estimate, such as the influence of temperature, pressure or scattering ratio on the Rayleigh response, could certainly contribute to improving the error characterisation. The ECMWF model equivalents of Rayleigh-clear are found to have a significantly better agreement with the radiosonde wind measurements compared to the Rayleigh-clear observations.

This is a further confirmation that the co-location parameters used for this validation study are appropriate and that the model equivalents provide a suitable reference for validating Aeolus. In addition, we demonstrate the existence of an orbital- and altitude-dependent bias in the Rayleigh-clear channel, which is visible with respect to both radiosondes and ECMWF model equivalents. This bias has already been documented by Borne et al. (2023) in West Africa using model equivalents and is now confirmed observationally. The underlying cause for this bias, however, remains unknown. In addition, we find that Rayleigh-

clear performs better compared to Rayleigh-cloudy, but due to the lack of Rayleigh-cloudy data we cannot draw any strong conclusions.

For Mie-cloudy, the statistical analysis yielded a systematic negative deviation of $-0.9 \pm 0.3$ ms$^{-1}$ within ESA specifications when uncertainty is taken into account, and it is consistent across all orbital nodes and Cal/Val sites. The random error between 2–16 km is $1.1 - 2.3$ ms$^{-1}$, which falls within the ESA recommendations. The general quality of Mie-cloudy winds does not



depend on the co-location radius, while it is more sensitive to temporal differences. The errors appear to be larger at 5 km and about 1 km altitude, typically at the upper and lower limits of the Saharan Air Layer, where clouds frequently occur. According to Lux et al. (2022a), the Mie fringe of the Fizeau interferometer can be distorted in the case of strong backscatter gradients, e.g. at cloud edges. Interestingly, Mie-cloudy does not seem to sample within dust layers, as most bins with high dust concentrations are rejected by the QC. Furthermore, the systematic and random Mie errors decrease with the percentage of cloud cover, while

they increase in the presence of dust. This may be attributed to the generally weak backscatter of dust, increasing the error of the Mie-cloudy winds. Similar to Rayleigh-clear, outliers with small error estimate and large absolute differences can be found for all co-location distance, co-location time, altitude, dust concentrations and cloud cover. An improvement of the Mie EE is expected from an optimisation of the Mie core algorithm, such as the fitting function or the classification algorithm.

    The presented study determined the error dependencies of the different Aeolus observation types and error estimates with respect to tropical clouds and dust. The acquired information are valuable to further improve the processing algorithms in order

to meet the requirements of the mission.

*Code availability.*    The analysis was conducted using the Python language and the code can be provided on request.

*Data availability.*    The Aeolus L2B and L1B products are provided by the European Space Agency (ESA) Earth Explorer Program and are available from the online Aeolus Data Dissemination Facility (https://aeolus-ds.eo.esa.int). The European Centre for Medium-Range Weather

Forecasts (ECMWF) model-equivalents are available in the ECMWF Meteorological Archival and Retrieval System (MARS) operational archive. Dust mixing ratio data is openly available and can be downloaded in the Copernicus Atmosphere Monitoring Service (CAMS) Data Store. The NW SAF Cloud Type (CT) product can by obtained using the software package NWC/PPS available from http://www.nwcsaf.org/. The radiosonde data corresponding to launches from Puerto Rico are publicly available at http://dx.doi.org/10.5067/CPEXAW/DATA101. The other radiosonde data can be provided on request.

*Author contributions.*    MB, PK and MW conceptualized the study and developed the methodology. MB, PK, MW and BW carried out the investigation and validation. PK, MW, PV and RR provided financial support for the project. MB, PV and RR were responsible for data curation. MB conducted the formal analysis and wrote the original draft of the paper. MB, PK, MW, BW, CF, PV and RR reviewed and edited the paper.

*Competing interests.*    The authors declare that they have no conflict of interest.



*Acknowledgements.* We would like to thank Thorsten Fehr for organising the Joint Aeolus Tropical Atlantic Campaign (JATAC), which made this cross-Atlantic study possible. We would also like to thank the German Aerospace Center (DLR), which assisted in the transport of radiosonde material and helium tanks from Germany to Sal in Cape Verde. Sincere gratitude also goes to the large teams involved in the radiosonde launches. For the Cape Verdean team, our special thanks go to Azusa Takeishi, Tanguy Jonville and Cedric Gacial for their hard work and dedication.



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
