# Peer review of "Validation of Aeolus L2B products over the tropical Atlantic using radiosondes"

_EGUsphere, 2023_

## Author Comment (AC3)

**Answer to Referee #2**

We greatly appreciate the insightful feedback provided by Referee #2, which we received on August 07, 2023. The comments from the referee that have been addressed in the manuscript are indicated in green, and the responses from the authors to the referee are highlighted in red.

This manuscript compares AEOLUS wind measurements with coordinated radiosonde observations, targeting the overpasses of AEOLUS, during a two month tropical field campaign. The comparison allows the AEOLUS observations to be placed within a proper uncertainty contex for future use by modelers and others. The paper is in mostly good shape, but there are a few problems that need to be fixed.

There is one major question unaddressed. The AEOLUS satellite has been in operation since 2018, so into its 5th year of measurements. With such a large data base it seems there should have been a number of near overpasses of the standard radiosonde network in the tropics during those 5 years. Is that not the case? If that is not the case it should be mentioned so the motivation for the dedicated radiosonde campaign is clear. If it is the case then such a study needs to be referenced, or the number of such previous coincidences needs to be mentioned and the reason for excluding them from this study. There is no mention of this in the literature review.

Thank you for your comment. We chose to focus on radiosonde data from the JATAC campaign for four specific reasons:

1. **Tropical Data Scarcity**: Radiosonde network in the tropics, especially the Global Observation System (GOS), has limited and irregular data due to infrequent reporting.
2. **Timing Gaps**: Radiosondes are usually launched at fixed times (12 and 00 UTC), creating observation gaps for the comparison with Aeolus, especially near the Greenwich meridian. Additional measurements at 06 UTC and 18 UTC were needed for comprehensive coverage.
3. **Challenges in Validation**: Changes in Aeolus data quality and product change over the past five years made a broad validation challenging. Focusing on the JATAC period allowed for a more accurate validation.
4. **Comprehensive Comparisons**: Restricting the comparisons to the JATAC period enables meaningful comparisons with other instruments used during the campaign, such as aircraft measurements conducted simultaneously.

To clarify our motivation, we have included the following sentence in line 75: "*Furthermore, our approach involves using radiosonde data exclusively from the JATAC campaign to facilitate more*

*comprehensive comparisons with other campaign instruments, considering the scarcity of radiosonde measurements in the tropics, the need for radiosonde launches at local dusk-dawn times to reduce timing gaps and possible variations in Aeolus data quality across different times and locations."*

The other major complaints concern the figures. First the choice of symbol/line size, color, and faintness is poor. Data in the figures should not be hard to see at normal zoom levels, and not hard to distinguish between one set of data and another, but presently that is the case. In some figures the lines indicating the data are practically invisible, and the colors chosen are too close to each other. Second there is no need to repeat in the text the figure captions. Leave that in the figure. In the text discuss the figure, the reader will find the figure caption.

Thank you for your feedback. We've addressed this concern and made the necessary adjustments to Figures 1, 2, 7, 8, and 9 based on the specific comments provided below and addressed in subsequent responses. We've also ensured that there's no redundancy between the captions and the main text throughout the manuscript. We believe these modifications have significantly improved the clarity of the figures, thanks to your valuable input.

Further specific comments on these issues and a few others, along with suggested corrections follow here by line number. Text in the manuscript, or corrections to that text, are set off with ellipses. While I am willing to review a second version, that is not necessary assuming the authors make a good faith effort to address these comments.

- General comments
    - 87 … with an angle of …
        - revised accordingly
    - 100 … respectively. The 87 km is required by the lower …
        - revised accordingly
    - 109 It would be helpful to briefly mention what particles are being observed for the Mie-clear observations. Later we find that Mie-clear is not used. Why introduce a classification that is not used for obvious reasons? Mie-clear must have no particles for scattering, so how can it work. Leave it out.
        - Thank you for your input. We have excluded "Mie-clear" from the text.
    - 112 Is this product identified by two numbers, 12 and L2bP 3.50? This is a bit confusing. Are both numbers important for the reader? We find later neither is used further.
        - A single mention of these two numbers is enough to provide precise information to the reader regarding the dataset utilized in this study.

- 113 It is not surprising that the Mie-clear signal is weak, which harkens back to line 109. This should be dealt with all at once. Not piecemeal. In addition there is a problem with this sentence related to the word "should".
  - revised accordingly
- 125-126 Does (4d-EnVar) have to be defined twice?
  - revised accordingly
- Table 1 – Of what importance is the weekday? More important would be the dates it seems. The times are very tight for Aeolus, usually a span of one minute. But is the orbit of Aeolus that stable that it would always be 50, or 180, or … km away from the sounding location on every profile on a given week day at exactly the same time? This needs explanation. It seems there would be some variability for soundings on different days, and some variability on the coincidence radius.
  - We omit listing all 20 dates, given their weekly recurrence. The weekday is just informative for this purpose. Our choice of a co-location radius is positioned closely along the satellite tracks but large enough to accommodate the orbit variability each week. Start and stop times have been rounded to the nearest minute, and they consistently exhibit similarities week after week.
- 135-136 KIT has already been defined, so use it. If an acronyms is not going to be used, don't define it. In fact I don't think KIT is used again.
- 137 Aren't all weather radiosondes light these days?
  - Yes indeed, "light" can be omitted.
- 141 Similarly, NASA has already been defined.
- 149 Don't all weather radiosondes provide wind speed, wind direction, temperature, humidity and air pressure?
  - Yes this information is indeed not necessary. We rephrased as following: "*The launches were coordinated by the KIT with local support from the JATAC team, using DFM-09 (GRAW) weather radiosondes.*"
- 171-173 Why is very high/high included for clouds below 7 km? Similarly if very high is for clouds above 16 km, why is it included for clouds between 7 and 16 km? Why are these classifications even mentioned? They are never used again.
  - Kindly refer to our response to the first reviewer.
- 182 …80 km and a time resolution of 6 hours …
- 192 What is meant by the radiosonde total horizontal wind speed? Isn't the radiosonde wind speed averaged over each Aeolus height bin?
  - When referring to total horizontal wind speed, we are indicating the combined magnitude of both wind components.
- 229 Generally ms-1 means per millisecond, whereas m/s is usually written as m s-1.
  - We changed this throughout the whole text.
- 244 … In contrast, the Mie …
- 247 How does a wind product get a validity flag of 0? Don't the authors just mean that, … all Aeolus wind products with EE above 8 ms–1 for Rayleigh and 4 ms–1 for

Mie, are omitted. … Why introduce a validity flag which is just a reflection of these criteria. The criteria mean something. The validity flag doesn't and is never mentioned again.

- In this section, we're describing the quality control procedure we implement before using the data for our study. The validity flag is an important part of this process, along with the EE thresholds, to remove observations that are not reliable and have been blacklisted. We don't revisit this aspect later because our primary objective here is to explain the initial data preprocessing steps.

- 267 Note that Mie-clear is not included here, which begs the question of why it was ever mentioned.
  - We have removed all references to Mie-clear in the text.

- 4.1.1 Comparative analysis with the ECMWF model equivalents – Isn't the comparison primarily between AEOLUS and the radiosondes? The ECMWF in the title is a bit confusing. It should be pointed out whether the ECMWF model equivalents have incorporated the sounding data, which was added to the GTS as mentioned earlier.
  - You're right, the title was somewhat misleading. Hence, we modified it to "Comparative analysis with radiosondes and ECMWF model equivalents". Additionally, in section 2.2 "Radiosondes", we already mentioned that "Most of the radiosondes launched at Sal were ingested into the Global Telecommunication System (GTS)". However, this detail isn't crucial to the main conclusion of this section. The key takeaway here is the strong agreement between radiosonde and model equivalents, highlighting the reliability of the co-location parameters used in this study.

- Figure 1. Don't use red and orange for two colors, particularly when the red has an orange tint. Use red and green or black, something that can be clearly distinguished. Make the symbols larger.
  - Thank you for your suggestions. We have replaced the orange color with green and increased the size of the symbols

- 269-274. Don't repeat the figure caption in the text. Let the figure caption do its job.
  - We appreciate your feedback. We have significantly condensed the description of the figure within the text.

- Table 3 In the caption introduce the quantities in the same order as they appear in the Table, as was done in Table 2, for consideration of the reader, not in the reverse order as done here.
  - We have revised the caption for Table 3 to maintain a consistent and logical order in the content description.

- Figure 2 and its caption need work.
- 1) The caption puts so many qualifiers in the first sentence that the reader is unsure what difference is shown. The caption should read something like. Differences between Aeolus (O) and radiosonde (RS) wind observations (dots) for a) Sal and b)

PR/SRCX for descending (blue) and ascending (red) profiles. The solid line and shading are the average differences and their standard deviations. Average differences with ECMWF model equivalents (B) are given as dotted lines. If this is correct, It is presently so confusing it is hard to be sure.

- We have revised the caption based on your suggestion to make it clearer and more readable. Thank you for your feedback!

- 2) The individual differences (dots) are too faint to be seen clearly.
    - We enhanced the data's visibility by increasing its opacity.

- 3) The dots and the averages and standard deviations are not consistent. The dots show much more spread than indicated by the standard deviations and are not consistent with the averages. For example, consider the descending profile in b) between 5 and 10 km. There are 2-3 blue dots below 0 m/s and 10 or more dots > 0 m/s, with a range of 2-8 m/s, yet the average line is between 0 and 2 m/s. Something is wrong.
    - The issue pointed out by both you and the other reviewer stemmed from one omission in the caption. We originally did not mention that we applied vertical smoothing on the line using a three-value moving average, which led to the confusion. To address this, we have incorporated an extra sentence into the caption.

- 4) It is not clear why the difference has to be multiplied by -1 for descending profiles. That just confuses the comparison, leaving the reader with the need to invert the descending profiles to compare with the ascending profiles. It is also not clear why this difference has to conform with the sign convention of the model coordinate system.
    - We use the model coordinate system for several reasons. Firstly, it offers a more intuitive means to represent the actual bias by displaying the actual wind difference between the two orbits. Furthermore, this choice facilitates comparisons with the study conducted by Borne et al., which documented this bias in West Africa and employed the model coordinate system. This observational validation of the bias serves as a valuable confirmation of the results observed previously in West Africa using model equivalents.

- 324-328 While the text does a somewhat better job than the figure caption there is still no need to repeat a figure caption in the text. Here is where the need to conform to the model convention and how this limits the ability to compare the ascending and descending profiles needs to be explained.
    - Thank you for your feedback. We have indeed shortened the figure caption in the text and included the following sentence to clarify the reason for adhering to the model convention: "*To better understand the variations in wind speed between the orbits and enable easy comparisons with studies like Borne et al. (2023), which has also documented this bias within the model coordinate system, we have adopted to the model sign convention. This involves multiplying the HLOS winds from descending tracks by -1.*"

- 328-329 So, considering the -1 multiplication of the descending differences, isn't this difference from -2.5 to 2.5 m/s? The fact that ascending and descending below 5 km both appear above 0 m/s in a) is a bit misleading? If they agree they should appear on opposite sides of 0 m/s as in b), correct?
    - Indeed, at 8 km altitude, the difference should be 5 m/s, not 2.5 m/s as previously mentioned. We've corrected this in the text. Below 5 kilometers over Sal, the presence of the Saharan Air Layer and dust particles could potentially contribute to the bias, but there is no hard evidence for this.
- Fig. 3 the cloudy colors are so faint as to be very difficult to see against the dominant blue. Use bright colors. If the Rayleigh-cloudy outlier symbols are the same size as the Rayleigh-clear outlier, then there are no such points on the plot. Don't include a legend for points that don't appear.
    - We improved visibility by changing the colors for Rayleigh-cloudy data to green and removed the outlier legend for Rayleigh-cloudy since they are not present in the figure.
- 422, 451, 460 Where are the transparent symbols? Are these the fainter symbols? This criteria is not defined in the figure caption, or in a legend on the figure, but should be. At present the reader does not know which symbols these are. There are already too many colors on the plot to distinguish a transparent symbol. How is transparent brown different than say yellow or orange? Use a different symbol mark: box, open circle, cross.
    - Thank you for your comment. We have updated the symbols for the faded values, replacing them with asterisks (*) to enhance their distinction from the other data points. Additionally, we have included the following sentence in the figure caption: The faint * symbols serve as references for values that did not meet the QC criteria.
- 450 What is a. u.? No panel in Fig. 6 has a scale extending to 5e13.
    - a.u stands for "arbitrary unit" as the useful signal is normalized. Thank you for pointing out the mistake in the exponent; the correct value is indeed 5e15.
- 458 kgkg-1 ?
    - The dust mixing ratio unit is in kg/kg.
- 464 Why show measurements which are artifacts and clearly wrong. Leave them out.
    - We present all measurements in accordance with the applied QC criteria, including those that were rejected (shown in a transparent format). This offers a chance to observe certain artifacts that are evidently incorrect but were not flagged by the QC process. This information is valuable for refining the QC procedure. We have added following sentence in the manuscript line 461: "*This highlights that some artifacts were not flagged correctly by the QC process.*"

- Fig. 7 Make the data visible! Use thicker lines. Use a darker gray so the reader can see all the data. There is a lot of space on the figure, don't make it difficult to see the data.
- Fig. 7e) legend …High semitransparent meanly thick clouds …? Do the authors mean mainly?
    - The name follows the SAF NWC product nomenclature. This cloud type is correctly referred to as "High semitransparent meanly thick clouds". To clarify this, we have added "The cloud names follow the SAF NWC product nomenclature." in the legend of Figure 7.
- 479-480 … it is not surprising … Rayleigh-cloud measurements ... Isn't this obvious? Surely this aspect of the algorithms have been clearly checked for assurance that clear sky conditions are determined.
    - Yes, this is obvious, so we have taken out "Rayleigh-cloudy" and "Mie-cloudy" from the sentence.
- 484 … with to a …?
    - Thank you for pointing out the error. We rectified the sentence by employing "with the".
- 489 In Table 1 the co-location radius was listed as 50 km, now it is 60 km. Recall the earlier comment about Table 1 and how the orbits could be so consistent with the co-location radii listed.
    - Thank you for noticing this mistake. You're right, it's 60 km in both cases. Kindly refer to our previous explanation in response to Table 1, where we elaborated on why we consistently selected these co-location radii.
- 514 … However in panel 9d, we see …
    - Thank you for pointing out the error that we have corrected.
- 532-533 Isn't this a little surprising. At large co-location radii there are going to be differences just due to geophysical variations over such a large distance. The atmosphere is not that homogeneous over distances that large.
    - We would argue that the atmospheric dynamics in the tropical Atlantic at this time of year are mainly driven by large-scale African Easterly Waves, which have a typical wavelength between 2000 and 4000 km. Furthermore, our results support this hypothesis as we don't see any error dependence with respect to the co-location distance.
- 554-556 Isn't this expected almost by definition. The signal is going to be cleaner without clouds so the instrument will perform better. Inherently Rayleigh-cloudy is going to give a weaker signal.
    - Yes, that's what we anticipated. Our aim here is to verify this expectation using the results of the validation study.

---

## Author Comment (AC4)

**Answer to Referee #1**

We greatly appreciate the insightful feedback provided by Referee #1, which we received on June 02, 2023. The comments from the referee that have been addressed in the manuscript are indicated in green, and the responses from the authors to the referee are highlighted in red.

The authors present statistics of the validation of Aeolus winds against independent ECMWF model fields and radiosondes. This is important work to gain knowledge on the errors of Aeolus winds. The region used for validation is limited to the tropics, which on the other hand is a very interesting region because of the challenging weather conditions with dust events and convective clouds and because of limited other Aeolus related Cal/Val campaigns in this region.

- Major comments

==============
- G1) At many places in the paper, the authors compare MADI against EEtot. This is a fundamental mistake as MADI is not a metric related to standard deviation such as EEtot (and SMAD). The authors can confirm this by taking a sample of random numbers, with normal (Gaussian) distribution, and compare the MADI value with the input standard deviation value.
    - Thank you for your comment. We acknowledge the fundamental error in our manuscript. We have therefore thoroughly removed all comparisons between MADI and EE throughout the document.
- G2) The authors should be more strong on their main conclusion in the abstract, by ending e.g. with: "Based on the data used in this study Aeolus Rayleigh winds do not meet the mission random error requirement and Mie winds do most likely not meet the mission bias requirement."
    - We acknowledge the need for a stronger conclusion to emphasize our findings. As you pointed out, the Mie bias doesn't meet the mission's random error requirement. However, when we account for the standard error of the bias, the Mie winds statistically align with the recommended value. Therefore, we have included the following sentence in the abstract of the manuscript: "*It is therefore concluded that Rayleigh-clear winds do not satisfy the random error requirement of the mission, whereas Mie-cloudy winds do so, when considering the standard error.*"
- G3) The classes discussed in lines 171 to 174 are very unclear. For instance, what is meant with "below 3 km (very high, high, mid-level, low, very low and fractional cloud types)"? How can you have very high clouds below 3 km? Also, why not using the useful signal at measurement level to identify clouds within the profile?
    - We recognize that the previous description of our classification method has caused confusion. The primary objective of this classification is to categorize each observation based on the presence of clouds along the satellite track.

The confusion stemmed from our earlier approach of listing cloud types both within and above each altitude range. To address this, we have only included cloud types within the specific altitude range. Moreover, we have added the following sentence to succinctly explain the classification's purpose: "According to this classification, an observation bin is considered as cloudy if it is situated within or below a cloud." We trust that this clarification provides a more straightforward and comprehensible explanation. In response to your second question, while using signal intensity can help us detect clouds in the atmosphere, it may not be accurate enough on its own to tell the difference between clouds and other particles, like dust.

- G4) line 195. Did you check this statement, e.g., using spectra following Skamarock (2008). They show that the area below the kinetic energy spectrum (which is actually the atmospheric variability over the integrated scales) can be quite substantial when starting at 340 km (or truncation wavenumber 60).
  - Indeed, the kinetic energy spectrum, as presented by Skamarock, exhibits significant fluctuations at scales smaller than 340 kilometers. Nevertheless, in the free troposphere, it is well-established that African Easterly Waves (AEWs) with scales of 2000 to 4000 kilometers are a majorsource of variability over West Africa. Consequently, we consider it appropriate to adopt a more flexible criteria for the colocation radii. Our findings confirm this hypothesis, as we observe no error dependency with respect to the colocation radii.

- General comments

================
  - line 11; measurements -> observations (Note that for Aeolus an observation is the result of accumulated measurements; mixing these terms in the text is confusing. Please correct everywhere in the text accordingly)
    - revised accordingly
  - line 15; the orbital-dependent bias of up to 2.5 m/s applies to only some parts of the atmosphere. This nuance should be made here.
    - revised accordingly
  - line 33; "..... along the LOS of the instrument, which is directed perpendicular to the direction of satellite propagation. Please add the last part.
    - revised accordingly
  - line 54: Replace: "..... that still needs to be explored ..... potentially affecting ....."
    => .... that needs further exploration .... which impact ....
    - revised accordingly

- line 117: replace ".... some SRs ....." by ".... small SR values, which are dominated by instrument noise, ...."
  - revised accordingly
- line 120; I do not understand what you mean with ".... and distances between the instruments and the height bins"? Please explain or rephrase.
  - We replaced "the instruments" by "Aeolus" to make this sentence clearer.
- line 123: ".... especially in the case of strong Mie returns, which are not detected by the classification procedure, ...." The addition is important because in principle measurements with strong returns should be classified as "cloudy" and not enter the Rayleigh-clear wind.
  - revised accordingly
- line 138-139; vertical resolution is not in m/s. Probably you mean that the balloon ascending speed is 5 m/s, then measuring every 2 seconds gives a vertical resolution of 10 m. Please correct.
  - revised accordingly
- line 216; this a surrogate for the standard error, Right? Please use this more well-known terminology in statistics, rather than "uncertainty of the mean bias".
  - Yes, we made the necessary corrections to the text by using the term "standard error".
- line 227; I guess the representativeness error is different for Mie and Rayleigh winds as they sample the atmosphere along different length scales, i.e., about 10-15 km for Mie and and about 90 km for Rayleigh, along the satellite track? Can the authors please comment on this?
  - Since identical co-location radii are used for Rayleigh and Mie, the observations from both channels are averaged using the same length scale. Although this averaging involves more data points from Mie compared to Rayleigh due to the differing integration lengths, we consider it reasonable to apply the same representativeness error range for both channels. We have included the following sentence at line 231: "*Note that despite the integration lengths differing, we average Rayleigh and Mie observations over the same co-location area, leading to the application of a consistent representativeness error range for both channels.*"
- line 239; with EE you mean EE_Aeolus as in Eq. (9), right? Please be consistent in the text
  - We did indeed mean EE_Aeolus. We checked the entire text for consistency and changed it accordingly.
- line 243; what do the authors mean with: "noise related to atmospheric temperature and pressure"? Do errors in these parameters lead to wind random errors or biases?
  - Rayleigh-clear winds are measured based on the double-edge technique, where the Doppler shift of the broadband molecular scattered light is measured by means of two Fabry-Perot interferometers that are spectrally shifted by several GHz. The ratio of the intensity measured behind these

Fabry-Perot interferometers is proportional to the wind speed. To retrieve the actual wind speed, calibration procedures have to be performed as discussed by Dabas et al. 2008 (Correcting winds measured with a Rayleigh Doppler lidar from pressure and temperature effects, Tellus A, 60, 206–215, 2008). As the shape of the spectrum of molecular scattered light (Rayleigh-Brillouin spectrum) changes with temperature and pressure, the calibration of the Aeolus Rayleigh channel depends on the accurate knowledge of these two parameters. Hence, any uncertainties in temperature and pressure contribute to the random error of the retrieved wind speeds. However, this uncertainty can be considered to be significantly smaller compared to the contribution of the SNR. For further clarification of this topic, we adapted the respective sentence in the paper manuscript according to: "*Future baseline versions are foreseen to also include contributions to the EE caused by uncertainties of NWP temperature and pressure used in the processor for instrument calibration procedures as well as the one caused by an insufficient correction of the narrowband particulate return that is transmitted to the Rayleigh channel (Dabas 2008)*".

- Figure 1. red and orange are hard to discriminate. Please use a different color for orange (Rayleigh-cloudy).
  - Thanks for your suggestion. We have switched the orange color to green.
- Caption of figure 1, please mention explicitly that you used model equivalents from the model background (which did not (yet) use the radiosonde), see also line 129. This is important, obviously and good to mention again.
  - Thank you for your comment. We have included this information in the caption.
- line 275 mentions a STD of 2.1 m/s for sqrt(<HLOS_ECMWF-HLOS_RS>^2) at Rayleigh-clear locations. The same metric shows a value of 2.93 m/s at Mie-cloudy locations in line 279. That is quite a large difference for parameters with quite consistent and well-known error characteristics. Assuming that the quality of radiosonde observations is rather constant for the complete profile this suggests that ECMWF performs substantially worse at locations where Mie winds are found (lower troposphere) than at locations of Rayleigh-clear winds (upper troposphere, lower stratosphere). Or is this discrepancy simply a statistical effect due to the limited data set? Can the authors please comment?
  - Thank you for raising this point. In simple terms, the ECMWF model equivalents perform less accurately in cloudy areas compared to clear-sky conditions. Since Mie-cloudy observations are mostly present in cloudy regions, it's expected that the model equivalents for these situations would show lower performance compared to the model equivalents from Rayleigh-clear observations, which occur in clear sky conditions. However, this result may also be influenced by the small statistical sample size of Mie-cloudy. We mentioned this point in the text, with the following sentence: "*Please note*

*that Mie-cloudy model equivalents, present in cloudy conditions, are anticipated to demonstrate lower performance compared to Rayleigh-clear model equivalents occurring in clear sky conditions.*"

- line 283; "as most of the systematic and random errors seem to be specific to the Aeolus Rayleigh-clear winds". But in the text above you show that Mie-cloudy biases are larger than for Rayleigh-clear. Please correct.
  - You are correct in pointing out that Mie has a more pronounced average bias, even though we have shown in section 4.1.3 that Rayleigh-clear exhibits an orbital-dependent bias. Therefore, we have removed the term "systematic" from the sentence in question.
- line 284: "This stresses the need to identify the underlying potential error sources of Rayleigh clear observations with respect to the presence of clouds and dust aerosols ......"
  - revised accordingly
- Given the larger systematic errors in Mie-cloudy I would think that these are more sensitive to clouds and aerosols. The fact that random errors are larger for Rayleigh than Mie is pretty clear. Please comment.
  - Please refer to our response to your earlier comment. We have addressed the issue by removing the systematic error in the sentence. You are correct in noting that when it comes to systematic errors, Mie-cloudy might be more influenced by clouds and aerosols.
- Table 2. sigma_mu is not defined in section 3.2. Please do.
  - We made an error by replacing "sigma_mu" with "epsilon_mu" without ensuring consistency throughout the text
- line 304; "For Mie-cloudy, the systematic difference indicates a bias of 0.9±0.3 ms1, which is within the uncertainty range of the ESA's specification ..." No, it is not, see major comment G2. Please correct.
  - Considering the standard error, the Mie error range does overlap with the recommended value of 0.7 m/s. This suggests that the Mie bias, accounting for the uncertainty represented by the standard error, is statistically consistent with the recommended range, when the standard error of the bias is taken into account. We have reformulated the text as follows, in line 309: "*For Mie-cloudy, the systematic difference reveals a bias of -0.9 ± 0.3 m/s, falling within ESA's specified uncertainty range when considering the standard error of the bias. This bias remains relatively consistent across regions and orbital nodes, with a slightly larger bias observed in the descending orbits and over Sal.*"
- line 306; how do you arrive at 1.1-2.3 m/s? Following Eq.8 with sigma_rep = 1.5-2.5, sigma_RS=0.7 and sigma_tot=2.9, I end up with sigma_Aeolus in the range 1.3-2.4. Where do I go wrong? See also table 3.
  - The results are not the same because you looked at information from Table 2, which includes data from all altitudes, while Table 3 only focuses on

heights between 2 and 16 kilometers for Mie-cloudy conditions. That's why the results are different. In the caption for Table 2, we have now specified "for all altitude ranges."

- line 315. I think AVATAR-T carries a 2 micron lidar, so measuring particles only. How can you compare these with Rayleigh-clear, measured in clean air conditions?

  - It is true that, besides the ALADIN airborne demonstrator (A2D), a 2-µm heterodyne detection wind lidar was flown onboard the DLR Falcon research aircraft during the AVATAR-T campaign. Due to the heterodyne measurement principle, the system indeed depends on the narrow-band particulate backscatter signal and will not provide winds from aerosol/particle-free atmospheric conditions. However, the detection scheme is much more sensitive compared to the direct-detection measurement principle used by ALADIN. Hence, the 2-µm wind lidar provides winds even at very low scattering ratios that are classified as Rayleigh-clear winds. This fact is for instance demonstrated by Fig. 3 in *Witschas et al., 2022*, which shows a flight example where the 2-µm wind lidar has almost full data coverage, and Aeolus measures almost only Rayleigh-clear winds. Hence, the 2-µm wind lidar is also well-suited to validate the quality of Rayleigh-clear winds. To clarify this fact, we added the following sentence to the paper: "*In this analysis, comparisons are only made with statistics derived from airborne wind lidar measurements acquired during the AVATAR-T campaign, which was also part of JATAC (Witschas et al., 2022; Lux et al., 2022b). In particular, the statistics derived from a heterodyne detection wind lidar (2-µm DWL) flown onboard the DLR Falcon research aircraft are used for comparison. Due to the high sensitivity of the heterodyne detection principle, the 2-µm DWL provides accurate wind speed data even in a clear atmosphere where Aeolus only provides Rayleigh-clear winds. Hence it is a well-suited reference instrument for the validation of both, Rayleigh-clear and Mie-cloudy winds.*"

- Figure 2. "Differences (dots) and average differences (lines)"
  I cannot conclude from the plot that the line is the average value. For instance in the left panel at 17500 m, all blue dots are on the right hand side of the line. Similar issues appear at all altitudes.

  - Thank you for your comment. We realized that we omitted mentioning in the caption that we smoothed the lines vertically using a 3-value moving average to reduce variability. We have made the necessary update to the caption, which now includes the following text: "*The lines were smoothed vertically using a three-value moving average.*"

- In the caption of Figure 2, mention Aeolus Rayleigh-clear winds.
- line 351; How is it possible to have '+'s with EE values > 5 m/s in figure 3a?

- In the caption, we mentioned that '+' symbols are defined for EE_Aeolus < 5 m/s. However, in Figure 3a, only EE_tot is shown, which explains values above 5 m/s.
- line 356; "discrepancy". This discrepancy is expected, see major comment G1.
  - Please refer to the response provided for the major comment.
- Figure 3. The binning of the stepwise solid lines is not explained. Why does it go up to 8 m/s in 3a, while you have much less than 40 data points at this value.
  - We acknowledge that the caption did not adequately describe the usage of the stepwise lines. Therefore, we have revised it as follows: "*The solid stepwise blue lines indicate the MADI, and the dotted blue lines represent the SMAD of Rayleigh-clear. Each step includes a minimum of 40 data points to ensure significance.*"
- line 407; "Table 4 describes the error dependency of the Rayleigh-clear observations with respect to the presence of clouds and dust"
  This classification is not clear. Do the authors mean presence inside the bin or from bins aloft or both?
  - We acknowledge that the previous classification caused confusion. This classification's purpose is to categorize each observation based on cloud cover along the satellite track, which is discussed mainly in section 4.2.2 "Cloud type and dust." In other words, when a bin falls within the 7 to 16 km range, all clouds within this range and those above 16 km are taken into account to calculate the percentage of cloud cover along the track. The confusion arose from listing cloud types both within and above each altitude range. To address this, we have changed the text to only specify the cloud types within each altitude range. The sentence "*According to this classification, an observation bin is considered as cloudy if it is situated within or below a cloud*" explains how this classification works.
- line 415-417. MADI compared against EE is invalid, see major comment G1. The conclusion that: "EEtot in clear sky conditions is well calibrated, while it is becoming gradually too low with the increasing presence of clouds and dust." is not well explained.
  - You are absolutely correct; we cannot directly compare MADI to EE. Consequently, we have eliminated all comparisons between MADI and EE in the manuscript. We have also revised the sentence as follows: "This underscores the trend where $EE_{tot}$ is slightly overestimated in clear sky conditions and gradually becomes underestimated with the increasing presence of clouds and dust.
- Table 4. In the caption replace 25% by 50%
  - revised accordingly
- Figure 7a. use a different x-axis scaling to better visualize the differences between the curves, e.g., x in [-25,35]. Also for fig 8a and 9a.

- - We chose this scaling to maintain consistency across all three figures (7a, 8a, 9a) while ensuring ample space for a legible legend. It represents the most suitable compromise we could achieve.
  - Figure 7b, how do you arrive at the blue curve? And how the grey curves? Are the latter obtained from Eq.(9) with EE_Aeolus from the L2B product?
    - Thank you for your comment. Indeed, the blue and grey curves are derived from Eq. 9 using EE_Aeolus data from the L2B product. We have clarified this in the caption, specifying that they were obtained from Eq. 9.
  - line 483; "with a minimum of 3.5 m/s ...". This value does not follow from fig 7b. Please correct.
    - Thank you for your comment. The value 3.5 m/s mentioned in the text was initially associated with EE_Aeolus, but we have rectified it to 4.2 m/s to align with EE_tot.
  - Figure 8; the grey lines in b/c/d in look the same as in figure 7. Same for Figure 9. Despite the completely different scenes. Where do these curves come from?
    - It is correct and intended that the grey lines look the same in figure 7, 8, 9, as they represent the lines of all the 20 radiosonde profiles, used as a reference. To avoid confusion, we slightly reformulated this part of the caption: "*(b) Vertical profile of the Rayleigh-clear $EE_{tot}$ (blue line), together with the $EE_{tot}$ of all 20 profiles (grey solid lines) obtained from Eq. \ref{eqn:EEtot} and their average (black solid line).*"

- Minor comments

  ==============
  - line 9; Raleigh -> Rayleigh
    - revised accordingly
  - line 11; can be degraded -> are degraded
    - revised accordingly
  - line 87; add "off-nadir" and "in the tropics" in "it points at 35 degrees off-nadir with an angle of ~10 degrees from the zonal direction in the tropics".
    - revised accordingly
  - line 109; processing chain -> mission
    - revised accordingly
  - line 112; L2bP 3.50 -> L2Bp version 3.50 (the rest of the paper uses L2B with all capitals)
    - revised accordingly
  - line 113; "should is"  - remove "should"
    - revised accordingly

- line 134; "Between the 7 and 28th ...". Correct to either: "Between 7 and 28 ...." or "Between the 7th and 28th of ....".
    - revised accordingly
- line 183; "in the presence of ...". Add "the"
    - revised accordingly
- line 200; remove "bin-to-bin"
    - revised accordingly
- line 209; Eq. (5) misses the index (i) between the brackets. Please correct.
    - revised accordingly
- line 232; "the the". Please correct
    - revised accordingly
- line 244; In contrary -> In contrast or Contrary? Please check.
    - revised accordingly
- line 473; black lines -> black line
    - revised accordingly

---

## Author Response (AR2)

**Answer to Referee #1**

We highly value the insightful feedback given by Referee #1 during the second revision. The comments from the referee that have been addressed in the manuscript are indicated in green, and the responses from the authors to the referee are highlighted in red.

Thank the authors for well considering my comments and suggestions in the updated manuscript. Some final points that need to be addressed before publication are the following

Thank you for your additional comments and suggestions. They have significantly contributed to clarifying the overall scope of the study and enhancing its quality.

- Specific comments

==============
  - 1) I am happy with the stronger statement made in the abstract now, in answer to my earlier comment G2, although it is incorrect. The new sentence: "It is therefore concluded that Rayleigh-clear winds do not satisfy the random error requirement of the mission, whereas Mie-cloudy winds do so, when considering the standard error." mentions random error only while my statement for Mie winds was related to the bias. Please correct.
  I agree that there is overlap between the mission bias requirement of 0.7 m/s and the Mie-cloudy bias of -0.9 +/- 0.3 m/s. However, I would disagree with the conclusion that: "Mie-cloudy winds fulfill the wind bias requirement, when considering the standard error". With this statistic, the probability of being within the mission requirement is much below 50%. I therefore recommend (again): "based on the data used in this study Mie winds do most likely not meet the mission bias requirement".
    - Thank you for your feedback. We acknowledge that, strictly speaking, Mie-cloudy winds do not meet the mission bias requirement. We revised the abstract sentence you referred to on line 11: "It is therefore concluded that Rayleigh-clear winds do not meet the mission's random error requirement, while Mie winds do most likely not fulfill the mission bias requirement."
  - 2) The authors state in their reply that: "the ECMWF model equivalents perform less accurately in cloudy areas compared to clear-sky conditions." Can the authors please provide evidence for this general statement by adding a reference in line 284?
    - Unfortunately, we couldn't find a reference to support this statement. Therefore, we decided to remove the following sentence: "Please note that Mie-cloudy model equivalents, present in cloudy conditions, are anticipated to demonstrate lower performance compared to Rayleigh-clear model equivalents occurring in clear sky conditions."

**Answer to Referee #2**

We highly value the insightful feedback given by Referee #1 during the second revision. The comments from the referee that have been addressed in the manuscript are indicated in green, and the responses from the authors to the referee are highlighted in red.

Considering the rather large deviations of the Aeolus wind estimates from the radiosonde data, e.g. Figs 1a), 2, 8a), 9a), and the relative tighter agreement of the Aeolus model equivalents, Figs 1b), 2, 8a), 9a), an alternative test of the usefulness of the Aeolus wind data would be to compare ECMWF model runs with, and without, incorporating the Aeolus wind data. Considering the complexity and expense of incorporating/not incorporating data into the model, and running the model for both cases, this may not be possible. But if it were, it would be a nice straight forward demonstration of the usefulness of Aeolus winds.

Thank you for your comment. Various Observation System Experiments have already been conducted at different weather centers to assess the utility of Aeolus data, and numerous papers on this topic are available. Most experiments have consistently demonstrated the added value of assimilating Aeolus data, especially in the tropics and the upper tropospheric region. One notable example is the impact observed in the ECMWF IFS, discussed in Rennie et al. 2021 ("The impact of Aeolus wind retrievals on ECMWF global weather forecasts").

Given the maturity of this paper and the quantitative/complexity of the error descriptions and discussions, it is surprising the number of places where inconsistencies still occur. The authors are urged to double check that each figure reference in support of a statement, does in fact support the statement, thereby building trust in the reader. Presently that is not the case. As before comments follow here by line number. Text in the manuscript, or corrections to that text, are set off with ellipses.

Thank you for your valuable feedback. Your insights have greatly contributed to enhancing the clarity of this manuscript by addressing inconsistencies and omissions.

- Specific comments

  ==============
  - 14-15 This sentence needs some help. Try: …Gross outliers, defined as large deviations from the radiosonde data, but with low error estimates, account for less than 5% of the data…
    - Thank you for the suggested sentence. We have made the necessary adjustments.

- 34 … which is directed perpendicular to the direction of satellite propagation… This direction could be downward or horizontal to the satellite propagation? Text on the next page suggests the latter. In any case it should be made clear for readers not intimately familiar with Aeolus.
    - Thank you for your feedback. We have revised the sentence for clarity as follows: "The instrument carries a direct detection Doppler wind lidar called ALADIN (Atmospheric LAser Doppler INstrument) that emits short ultraviolet (UV) pulses at 355 nm along the Line Of Sight (LOS) of the instrument, perpendicular to the satellite's ground track and oriented 35° off-nadir."
- 2.2 Radiosondes. Only about ¼ of the radiosondes flown during the campaign matched Aeolus overflights. Why so few and why were the other ¾ of radiosondes flown? Twenty comparisons is not that many for the length of the campaign. How does number compare to the number of Aeolus overflights of each station during the campaign
    - The JATAC campaign over Sal lasted approximately three weeks, during which only co-located radiosonde profiles were utilized—corresponding to only three Aeolus overpasses per week in Sal for comparison. In regions like Puerto Rico and Saint Croix, radiosondes were not consistently co-located with Aeolus in space and time, as it wasn't the primary focus of the CPEX-AW campaign, resulting in fewer available comparisons. The additional radiosondes launched in Cape Verde were added to the Global Observing System for operational assimilation in NWP centers. This dataset also contributes to various studies, including comparisons with GNSS data within the framework of the CADDIWA campaign.
- Figure 2. Even with the three value moving average it is difficult to reconcile the individual data, points, with the O-RS lines and their standard error. Presuming the line to represent the mean it is difficult to reconcile the ascending line in Fig. 2b) with the corresponding data points. Concentrating only on the first 3 levels. Level 1 all data are > than the mean at -5 m/s. Level 2 only one data point is below the mean at -3 m/s, whereas 4 data points are well above that. Finally at level 3 the mean and data make sense. It is not clear how a 3 value moving average will produce the line represented for these three levels.
    - Thank you for your input. To utilize the moving average function, the "Minimum number of observations in window required to have a value" is set to 2 at both ends of the profile. Although this may impact the expected outcome at these points, our code verification revealed no errors, and the average values align with the individual data points. However, for clarity and to prevent confusion, we have opted to exclude the data points and retain only the vertical average of the points in the figure.
- Figure 3, the d) label is not defined in the caption. At normal zoom levels it is very difficult to see the Rayleigh-cloudy data. Use a clearly distinguishable color, e.g. red, or cyan. The green is so dark it appears black.

- Thank you for your comment. We have added the omitted (d) in the legend. Initially, our intention was to assign a specific color to each type of observation throughout the manuscript. We initially chose orange, but due to its close resemblance to the red used for Mie-cloudy, we switched to green as suggested by the referee. Subsequently, we now opted for the golden color, as cyan appears too clear and red is already used for Mie-cloudy. We made consistent updates to all figures (1, 2 and 8) and the text.

- 386 … outliers as values exceeding an absolute error of 6 m s–1 along with EEs inferior to 3 m s–1… According to the data in Fig. 4a) the outliers are at EEtot < 3.6 m/s, not < 3 m/s.
  - EEtot denotes the total error estimate, encompassing both radiosonde observation error and representativeness error. On the other hand, EE specifically pertains to the error estimate attributed to Aeolus alone. The distinction between 3 m/s and 3.6 m/s primarily stems from the conversion between EE and EEtot as presented in the figures.

- Table 4 caption …for dust mixing ratios above (Dust) and below (DustNO) 109 kgkg–1 along the track… For the readers sake, organize this sentence, here and in the text, to follow the columns in the table, as is done for cloudiness, not the reverse. Even better use this … for dust mixing ratios < (>) 10^9 kg kg^-1 DustNO (Dust) along the track.
  - Thank you for your feedback. Following your suggestion, we have revised the caption to enhance clarity. It now reads: "This comprises three categories of cloud cover (< 50%, > 50%, > 75%) and two dust mixing ratio sub-categories (> $10^8$ (Dust), < $10^8$ (noDust)) kg/kg along the track."

- 415 Why are the dust criteria different in the text and table caption. Perhaps the text is correct?
  - Thank you for bringing this to our attention. In the course of correcting the dust criteria from $10^9$ to $10^8$ kg/kg in the text during the first revision, we overlooked updating the caption. We have now rectified this oversight.

- 460 …quality as well, with an average MADI of 1.5 m s–1 (Fig. 4b)… Where does 1.5 come from? In Table 5 it is 1.6 at best, while Fig. 4b) shows it be more, primarily near 2.3.
  - Thank you for bringing this to our attention. The mention of 1.5 m/s corresponds to values above 12.5 km altitude, which unfortunately are not depicted in Figure 4b due to the limited size of the dataset. To prevent confusion, we have removed this sentence from the text.

- 467 …Surprisingly, observations sampled at the lower 1 km have the lowest normalized useful signals, mostly below 5e15 a.u. and are not discarded… Where are these observations? All accepted observations, except one, have a signal above 0.5 e 14 a.u., Fig. 6g).

- Yes, there is just one outlier below this threshold. After thoughtful consideration, we have chosen to omit the description below 1 km, as we believe discussing it is not crucial within the context of this paper.
- General comment on the color scales and text. Why use scales from 0 – 1.0 in the figure axes and color scales, and then refer to them in the text in the realm of 1 – 10, forcing the reader to do the conversion from text to figure? Make it simple for the reader, use the same scale in the figure as the one discussed in the text, or in the text discuss in the same units as in the figure.
  - We assume you are addressing the dust concentration colorbar. We chose a 1-10 colorbar for better visibility of gradients, having found it to be the most suitable after trying different options. In the text, our focus is on whether values are above or below a specific threshold, so there's no need to investigate into individual categories within the 1-10 range.
- 469 …They also correspond to the largest MADI scaling up to 4 m s–1 on average (i.e. Fig. 4d)… This statement is not consistent with Fig. 4d) where only 3 out of many observations are > 4 m s-1. The majority are well less than that.
  - Indeed, you've identified errors that resulted from corrections omitted during the first revision. As mentioned in the previous comment, we have now removed this part from the manuscript.
- Figs 7-9. The source of the 20 profiles mentioned in the caption to Fig. 7, …together with the EEtot of all 20 profiles (grey solid lines)…, is not explained and is not clear to this reader. Does Aeolus make 20 profiles within the coincidence criteria, or? What determines the blue one that is used?
  - The 20 profiles refer to those meeting the co-location criteria outlined in section 3.1 and specified in Table 1. To prevent confusion, we have included "described in Table 1" in the caption of Fig. 7. The blue line in figures 7, 8, and 9 corresponds, respectively, to three selected case studies, each characterized by distinct clear-sky, cloudy, and dusty conditions. The first sentence in section 4.2.3 is highlighting this.
- 498 … which corresponds to SAFNWC CT at 07:30 UTC, Aeolus overpasses a variety of high clouds, mainly high semitransparent clouds… According to the color scale explanation in Fig 8e) Aeolus passes through "High semitransparent meanly thick clouds". First, the statement in the text is not consistent with the figure scale. Second, what is a "meanly thick cloud"? Third, isn't "semitransparent … thick" clouds an oxymoron
  - Firstly, in the text, our focus is on the vertical location of clouds, which is why we specifically mention "high clouds." The exact cloud types is referenced in the figure using the NWSAF nomenclature, but it is not crucial for the discussion and, therefore, omitted. Secondly, detailed descriptions of cloud types as per the NWSAF classification can be found at https://www.nwcsaf.org/ct_description. This classification distinguishes between meanly thick and high semitransparent thick clouds, which is "based

on the use of CMA and spectral & textural features computed from the multispectral satellite images and compared with a set of thresholds.".Thirdly, we acknowledge that it may seem like an oxymoron, but we are just adhering to the nomenclature provided by the NWSAF product.

- Why are the spatial scales used for Figs 7e), 8e), and 9e) so vastly different?
    - The spatial scales differ due to the application of different co-location radii for the three cases, as detailed in Table 1 and in section 3.1.